



# PM$_{2.5}$ Assimilation within JEDI for NOAA's Regional Air Quality Model (AQMv7): Application to the September 2020 Western U.S. Wildfires

Hongli Wang[1,2], Cory Martin[3], Jérôme Barré[4,5], Ruifang Li[1,2], Steve Weygandt[2], Jianping Huang[6], Youhua Tang[6,7], Hyundeok Choi[8,3], Andrew Tangborn[8,3], Kai Wang[9,3], Haixia Liu[9,3], Jeffrey Lee[10]

1. Cooperative Institute for Research In Environmental Sciences, University of Colorado, Boulder, CO 80305

2. NOAA Global Systems Laboratory, Boulder, CO 80305

3. NOAA/NWS/NCEP/EMC, College Park, MD 20740

4. NASA Global Modeling and Assimilation Office, Greenbelt, MD, USA

5. Morgan State University, Baltimore, MD, USA

6. Center for Spatial Information Science and Systems, George Mason University, Fairfax, VA 22030

7. NOAA Air Resources Laboratory (ARL), College Park, MD 20740

8. SAIC@NOAA/NWS/NCEP/EMC, College Park, MD 20740

9. LINKER@NOAA/NWS/NCEP/EMC, College Park, MD 20740

10. School of Meteorology, University of Oklahoma, Norman, OK 73072

*Correspondence to*: Hongli Wang (hongli.wang@noaa.gov)

**Abstract.** This paper describes efforts to establish aerosol data assimilation capabilities for NOAA's National Air Quality Forecasting Capability (NAQFC), a regional online air quality modeling (AQM) system under NOAA's Unified Forecast System (UFS), by assimilating measurements of fine particulate matter (PM$_{2.5}$, particles with diameters less than 2.5 μm). PM$_{2.5}$ assimilation is developed within the Joint Effort for Data assimilation Integration (JEDI) framework and tested using its 3D-Var data assimilation (DA) component. The PM$_{2.5}$ observation operator is constructed by combining newly developed PM$_{2.5}$ transformation recipes in the JEDI Variable Derivation Repository (VADER) with a general spatial interpolation operator in the Unified Forward Operator (UFO).

Cycled DA and forecast experiments were conducted from 1 to 21 September 2020, during a period of Western U.S. wildfires, to assess the impact of assimilating PM$_{2.5}$ observations from the AirNow and PurpleAir networks. The control and analysis variables include individual aerosol species, with background error standard deviations generated by scaling their respective background values. Prognostic variables such as aerosol particle number and total particulate surface area are updated accordingly following each analysis update. All DA experiments use a 3-hourly cycling interval, with PM$_{2.5}$ observations assimilated every 3 hours. The control experiment uses the same configuration but without any data assimilation. Results show that assimilating either AirNow or PurpleAir PM$_{2.5}$ data reduces 1–24 h forecast errors in terms of mean absolute error (MAE) and root mean square error (RMSE)





compared to the control run over CONUS. Forecast skill, measured using the Critical Success Index (CSI) for $PM_{2.5}$
thresholds of 5, 12, and 35 µg/m³, also improves. AirNow observations have a greater impact overall, while
PurpleAir shows its strongest impact over Nevada, northern Utah, Colorado, and southwestern New
Mexico—regions with persistent underpredictions in the control run at forecast hour 1. Overall, the assimilation of
PurpleAir observations in addition to AirNow data leads to a slight reduction in 3–24 h MAE.

## 39  1 Introduction

Particulate matter with an aerodynamic diameter of 2.5 micrometers or smaller ($PM_{2.5}$) is a major contributor to
poor air quality in the United States, posing significant risks to public health and the environment, and contributing
to substantial loss of life. Over the past few decades, poor air quality in the U.S. has contributed to over 100,000
premature deaths annually, far exceeding fatalities from all other weather-related causes combined, which average
around 500 per year (Huang et al., 2025). Given its public health significance, $PM_{2.5}$ is one of the primary pollutants
used in calculating the Air Quality Index (AQI)—a standardized system designed to communicate daily air pollution
levels to the public. Elevated $PM_{2.5}$ concentrations frequently result in "unhealthy" AQI ratings, triggering health
advisories and public warnings.
$PM_{2.5}$ in the United States originates from a range of both anthropogenic and natural sources. Anthropogenic
sources include agricultural activities and combustion processes, such as emissions from motor vehicles, power
plants, industrial facilities, and residential heating systems. Among natural sources, wildfires are a particularly
significant contributor, especially in the western United States, where their frequency and intensity have escalated
dramatically over the past two decades (Wen and Burke, 2021). According to the U.S. Environmental Protection
Agency (EPA), wildfires account for approximately 15% to 30% of total $PM_{2.5}$ emissions nationwide (EPA, 2017).
While national seasonal averages of $PM_{2.5}$ have generally declined, summer $PM_{2.5}$ concentrations in the western
U.S. have remained persistently high, primarily due to wildfire smoke (O'Dell et al., 2019). In addition to degrading
air quality, wildfires have caused widespread property loss. Since 2005, more than 99,500 homes, businesses, and
other structures have been destroyed by wildfire-related events
(https://headwaterseconomics.org/natural-hazards/structures-destroyed-by-wildfire, last access on June 30, 2025),
underscoring the urgent need for more effective strategies in air quality monitoring, forecasting, and wildfire
management.
The National Oceanic and Atmospheric Administration (NOAA) has developed an advanced regional Air Quality
Modeling (AQM) prediction system within the Unified Forecast System (UFS) framework to enhance the accuracy
of air quality forecasts across the United States, particularly during wildfire events (Huang et al. 2025). The National
Air Quality Forecast Capability (NAQFC), operated by NOAA's National Weather Service (NWS), has been
providing operational air quality forecast guidance for over 20 years, with continuous inclusion of new capabilities.
Under NAQFC, the AQM version 7 was implemented and became operational on May 14, 2024. A key innovation
in this system is the integration of the Real-time Aerosol and fire behavior Visual Estimator (RAVE) — a high



spatiotemporal resolution, satellite-derived wildfire product — which enables a more accurate representation of
wildfire emissions. The system also features online coupling of atmospheric and chemical models, allowing
dynamic interactions between meteorology and atmospheric chemistry. This integration improves the representation
of emissions and ensures real-time feedback of meteorological fields that influence chemical transformations and the
transport of pollutants in the atmosphere. The UFS-AQM online system has consistently shown improved
performance in simulating major wildfire events, including the significant wildfires in the northwestern coastal
regions of the U.S. in September 2020, and widespread smoke transport from Canadian wildfires in the summer of
2023. This system was officially implemented on May 14, 2024 as NOAA's operational air quality prediction system
(AQMv7), replacing the previous offline-coupled  the Global Forecast System using the Finite Volume Cube-Sphere
dynamical core (GFS-FV3) version 15 with the Community Multiscale Air Quality modeling system (CMAQv5.0.2)
modeling system. (Chen et al. 2021).
$PM_{2.5}$ data assimilation (DA) has proven effective in reducing errors in air quality forecasts (e.g., Pagowski et al.
2010, 2012; Schwartz et al. 2012; Wu et al. 2015; Robichaud 2017; Lee et al. 2021; Chen et al. 2022, Ha 2022;
Vogel et al. 2025, among others). Pagowski et al. (2010) demonstrated that fine aerosol forecasts benefit from
AirNow $PM_{2.5}$ DA, showing improved verification scores for a period of at least 24 hours. Schwartz et al. (2012)
found that assimilating AirNow $PM_{2.5}$ observations significantly improved surface $PM_{2.5}$ forecasts over the
CONUS compared to forecasts without DA. Wu et al. (2015) reported that incorporating ground-based $PM_{2.5}$
observations notably enhanced 24-hour forecasts during a severe pollution episode in Shanghai. Similarly, Chen et
al. (2022) showed that assimilating multi-source $PM_{2.5}$ data significantly improved WRF-Chem $PM_{2.5}$ forecasts
with benefits lasting up to 48 hours. Lee et al. (2021) highlighted the effectiveness of assimilating ground in-situ
surface PM2.5 observations in improving the short-term PM2.5 predictions in Northeast Asia.
Many operational regional air quality prediction systems around the world use some form of data assimilation to
initialize the forecasts. These approaches vary in complexity, ranging from simple optimal interpolation to full
variational or ensemble Kalman filter methods  (e.g. Robichaud et al. 2016; Wei et al. 2024; Colette et al. 2024). In
NOAA's current regional air quality model (AQM) operations, aerosol and chemical initial conditions are
"warm-started" using 6-hour forecasts from the previous model cycle. The implementation of an aerosol data
assimilation system can further enhance short-term air quality forecasts by providing more accurate spatial analyses
of initial aerosol distributions.
To establish aerosol data assimilation capabilities for NOAA's regional operational AQM system, we employ the
Joint Effort for Data assimilation Integration (JEDI) (Trémolet and Auligné, 2020). JEDI is a flexible, agnostic, and
modern data assimilation system applicable to a wide range of forecasting systems (e.g. Liu et al. 2023; Huang et al.
2023; Sluka, 2024). JEDI offers a platform that supports efficient scientific development and facilitates the transition
from research to operations. As part of a broader strategic shift, NOAA and partner agencies are transitioning their
data assimilation systems to JEDI, opening the door for rapid integration of new scientific advancements, greater





consistency across modeling systems, and enhanced collaboration across research communities and operational
centers.
This study aims to develop an initial aerosol analysis capability for the AQM system by assimilating $PM_{2.5}$
observations using the JEDI 3D-Var framework. Low-cost PurpleAir data are valuable for real-time air quality
monitoring and are displayed in the AirNow Fire and Smoke Map (https://fire.airnow.gov/, last access: July 19
2025). However, their impact on numerical air quality prediction has not been thoroughly studied. In addition to
AirNow $PM_{2.5}$ measurements, this study also evaluates the impact of assimilating PurpleAir observations.
The paper is organized as follows: section 2 provides a description of Methodology including the NOAA's AQM
system, 3D-Var approach, and JEDI $PM_{2.5}$ assimilation. Experimental setup is presented in section 3 including case
description, AQM configuration, AirNow and PurpleAir $PM_{2.5}$ observations and background errors setup. Results
are described in section 4. A summary and conclusion are presented in the final section.

## 2 Methodology

### 2.1 AQMv7 overview

The NOAA's regional operational AQMv7 system was developed through the online coupling of the Finite-Volume
version 3 (FV3) dynamical core -based atmospheric model (Black et al 2021) with the EPA's Community Multiscale
Air Quality (CMAQ) model v5.2.0 within the UFS framework (Huang et al., 2025). In this UFS-AQM online
system, CMAQ is treated as an atmospheric chemistry column model to simulate atmospheric chemistry reactions
that govern concentrations of chemical species including gas- and aerosol-phase species. The transport terms of
chemical species are handled by the FV3 dynamical core in the same way as other physics tracers (Huang et al.,
2025). Aerosol module version 6 (AERO6) (Zhang et al. 2018) is utilized by CMAQ to simulate aerosol processes.
The AQMv7 system is configured over the North American domain with a grid-spacing of 13 km and 65 vertical
levels, extending up to 0.2 hPa. In total, AERO6 simulates 76 aerosol-related variables. Additional information
about the UFS-AQM online system can be found in Huang et al. (2025). In this research, the model configuration is
the same as the operational AQMv7 setup except for running over the CONUS domain with a 3 hourly cycling
interval.

### 2.2 $PM_{2.5}$ assimilation within JEDI 3D-Var

In the JEDI framework, a series of components are provided to create a flexible, comprehensive data assimilation
system. The JEDI three-dimensional variational (3D-Var) component is used to assimilate PM2.5 for AQMv7. The
3D-Var method is chosen for its operational feasibility, primarily due to its low computational cost and the fact that
it does not require an ensemble prediction system.





In practice, a 3D-Var data assimilation system typically uses an incremental approach to minimize a quadratic cost
function which is defined in terms of the analysis increment $\delta x$ relative to the guess state $x_g$:
$J(\delta x) = \frac{1}{2}(\delta x - \delta x_g)B^{-1}(\delta x - \delta x_g)^T + \frac{1}{2}(\mathbf{H}[\delta x] - d)R^{-1}(\mathbf{H}[\delta x] - d)^T$  (1)
Where:
- $\delta x_g = x_b - x_g$ is the guess state departure from background state $x_b$, which is usually taken from a
previous short-term forecast.
- $\mathbf{H}$ is the linearized observation operator of nonlinear observation operator $H$.
- $\mathbf{B}$ and $\mathbf{R}$ are the background and observation error covariance matrices, respectively.
- $d$ is the innovation vector, defined as:
$d = y - H(x_g)$  (2)
with $y$ representing the observation vector.
Once the increment $\delta x$ is obtained, the analysis state $x^a$ is reconstructed as:
$x^a = x_g + \delta x$  (3)
**2.2.1 PM$_{2.5}$ observation operator**
In AQMv7, the modal approach taken in the CMAQ model represents aerosol particle size distributions as the
superposition of three lognormal modes: Aitken (I), accumulation (J), and coarse (K). It predicts only three integral
properties of the size distribution for each mode: the total particle number concentration, the total surface area
concentration, and the total mass concentration of the individual chemical components.
The total PM$_{2.5}$ concentration is calculated as a weighted sum of the individual aerosol concentration across these
three modes:
PM$_{2.5}$=ATOTI·PM25AT+ATOTJ·PM25AC+ATOTK·PM25CO  (4)
Here, ATOTI, ATOTJ, and ATOTK represent the total aerosol mass concentrations in the Aitken, accumulation, and
coarse modes, respectively. For example, ATOTI is the combined mass of 14 prognostic aerosol variables in the
Aitken mode from the AERO6 aerosol module. Similarly, ATOTJ and ATOTK are the aggregated mass
concentrations of 49 and 7 aerosol variables in the accumulation and coarse modes, respectively. PM25AT,
PM25AC, and PM25CO are mass scaling factors for the three modes that vary by location and time. The aerosol
variables within the same mode share the same mass scaling factor.
The PM$_{2.5}$ observation operator is constructed by combining PM$_{2.5}$ transformation recipes in the JEDI Variable





Derivation Repository (VADER) with a general spatial interpolation operator in the Unified Forward Operator
(UFO). VADER is responsible for transforming model variables using user-defined "recipes" to generate new
variables in model space. For $PM_{2.5}$ assimilation, VADER computes $PM_{2.5}$ from individual aerosol species using
model-specific transformation, specifically using the equation 4 for this application. Since $PM_{2.5}$ composition varies
by model, these transforms are implemented within VADER to match the specific structure of the regional air
quality model AQMv7. Once $PM_{2.5}$ is derived in model space, UFO applies a generic spatial interpolation operator
to map the model-simulated values to the observation locations, enabling computation of the observed minus
forecast values.
The input for the $PM_{2.5}$ transformation are mixing ratio of the 70 aerosol variables wrt dry air in unit ug/kg, the
three mass scaling factors in the three modes, and dry air density for unit conversion. The output product is the
$PM_{2.5}$ in unit ug/m$^3$. It is noted that a recipe that uses temperature, surface pressure, and delta pressure to derive the
dry air density in case the dry air density is not found in the input variable list into VADER.
The JEDI/VADER $PM_{2.5}$ recipe provides nonlinear (NL), tangent linear (TL), and adjoint (AD) transforms of $PM_{2.5}$
that keeps the output products in the same grid space as the input variables. Hence, the generic interpolation operator
in UFO is used to connect the model-derived 3D $PM_{2.5}$ fields with observed surface $PM_{2.5}$ measurements. This
respects the JEDI paradigm of keeping the UFO part of the JEDI model independent.
**2.2.2 Background error covariance modeling**
In a 3D-Var system, the background error covariance (BEC) determines both the spatial spreading of information
from observations and the magnitude of the analysis increments along with the observation error variance.
The background error covariance matrix B can be decomposed into a standard deviation matrix ($\Sigma$) and a correlation
matrix (C), as follows:
$B = \Sigma C \Sigma$ (5)
The correlation matrix C is generally non-diagonal. $\Sigma$ is a diagonal matrix, with the standard deviations of the
background errors for each variable on the diagonal.
The error modeling of the correlation matrix and standard deviations usually apply to control variables. In the first
implementation of aerosol data assimilation in JEDI for AQMv7, the control variables are defined as individual
forecast aerosol variables, resulting in 70 control variables for AQMv7 with the AERO6 aerosol mechanism. The
setup of background error standard deviation and correlation modeling will be described in Section 3: Experimental
setup.
**2.2.3 Minimization Algorithm (DRIPCG)**



JEDI provides several minimization algorithm options. In this paper, we use the Derber–Rosati Inexact
Preconditioned Conjugate Gradient (DRIPCG) algorithm (Derber and Rosati, 1989), as implemented in the JEDI's
OOPS (Object-Oriented Prediction System) framework. DRIPCG has been extensively tested and is chosen here for
stability and convergence efficiency.

## 3 Experimental setup

### 3.1 The September 2020 fire event

The wildfires of September 2020 ranked among the most intense in the U.S. in recent years. These fires produced
dense smoke that initially moved westward over the Willamette Valley and eventually blanketed the broader region.
As a result, air quality rapidly deteriorated to hazardous levels, marking one of the worst air quality periods in recent
decades (Mass et al., 2021). Wildfire smoke originating from California, Oregon, and Washington was injected into
the free troposphere and transported across the country by prevailing winds, leading to hazy conditions in several
states. According to Li et al. (2021), from August to October 2020, wildfires in the western U.S. contributed 23% of
surface $PM_{2.5}$ across the contiguous United States (CONUS), with higher contributions observed along the Pacific
Coast (43%) and in the Mountain Region (42%). This study focuses on the peak fire activity occurring between
September 1 and 21.

### 3.2 $PM_{2.5}$ observations

In this study, surface $PM_{2.5}$ observations were obtained from two sources: AirNow and PurpleAir observing
networks. These datasets differ in sensor type, spatial coverage, and quality control (QC) requirements. AirNow
provides regulatory-grade measurements from federal, state, and local monitoring stations, while PurpleAir is a
low-cost, community-based network of air quality sensors. PurpleAir sensors are widely deployed by individuals
and communities, providing real-time data on $PM_{2.5}$ concentrations as well as meteorological variables such as
temperature, pressure, and relative humidity. Only the data reported from outdoor PM2.5 sensors are used in this
study. The PurpleAir data were available for registered users through the  PurpleAir API.
(https://community.purpleair.com/t/api-use-guidelines/1589)

### 3.2.1 PurpleAir $PM_{2.5}$ quality control and correction

Quality control and correction of PurpleAir data followed the methodology described in Barkjohn et al. (2021).
Readers are referred to that paper for further details. A correction is required because the PurpleAir raw data usually
overestimate PM2.5 concentrations under typical ambient and smoke-impacted conditions. The following quality
control (QC) filters were applied to the raw PurpleAir $PM_{2.5}$ measurements:
●   Reported $PM_{2.5}$ values from  two Plantower sensors within the PurpleAir sensor (channels A and B) must

be nonnegative.





• The PurpleAir sensor channel A and B consistency:
○ Absolute difference < 5 μg/m³, *or*
○ Relative difference within 61%.
• PM$_{2.5}$ values must not exceed PM10 values.
• PM$_{2.5}$ values must be less than 3000 μg/m³ (upper threshold).
• Gross check of relative humidity with range 0-100%.
Only PurpleAir PM$_{2.5}$ measurements that passed all the above QC criteria were retained for subsequent correction.
**3.2.2 PurpleAir PM$_{2.5}$ correction**
Correction of PurpleAir PM$_{2.5}$ measurements was performed using a multiple linear regression model based on
sensor-reported PM$_{2.5}$ (PA) and relative humidity (RH), following the correction formula proposed by Barkjohn et
al. (2021):
$$PM_{2.5}=0.524×PA−0.0862×RH+5.75 \tag{6}$$
We adopt the above equation because it was United States-wide valid by fitting data from September 2017 until
January 2020. Though the above correction equation is originally for 24h averaged PM$_{2.5}$, a similar regression
equation was derived from the September 2020 1h averaged PM$_{2.5}$ dataset:
$$PM_{2.5}=0.508×PA−0.0449×RH+4.89 \tag{7}$$
The close similarity between the two equations supports the consistency and robustness of the correction method
across datasets and time periods.
To reduce random sensor noise and improve comparability with the model resolution (~13 km), the corrected
PurpleAir PM$_{2.5}$ data were spatially averaged onto a 0.1° × 0.1° latitude–longitude grid. PurpleAir shows a good
coverage of Washington, Oregon, California and Colorado, and more observations of Arizona, Utah, New Mexico,
Texas.
**3.2.3 Observation error assignment**
Observation error standard deviations were assigned to each network:
• AirNow PM$_{2.5}$: 5% of observed value
• PurpleAir PM$_{2.5}$: 10% of observed value
The values above are based on the EPA's definition of acceptable measurement uncertainty, which specifies a 10%
coefficient of variation for total precision. The AirNow PM$_{2.5}$ observation errors were set to 5% of the observed
values. Park et al. (2022) also used a 5% error specification for assimilating PM$_{2.5}$ observations, though their study





focused on observation networks over China and Korea. For PurpleAir PM$_{2.5}$ data, the observation errors were set
to 10%, reflecting the higher likelihood of greater uncertainties associated with lower-cost sensors. For comparison,
the default PM$_{2.5}$ observation error configuration in Gridpoint Statistical Interpolation (GSI) (Pagowski et al., 2012;
Wang et al., 2021) includes a measurement error modeled as $1.5 + 0.75\% \times$ PM$_{2.5}$, along with a representativeness
error component. At the current model resolution of 13 km, the error specification used in this study reduces the
influence of large PM$_{2.5}$ observations, particularly those exceeding approximately 55 μg/m³ for AirNow monitors
and 25 μg/m³ for PurpleAir sensors.
Figure 1a–b shows the spatial distribution of AirNow and PurpleAir PM$_{2.5}$ monitoring stations at 1200 UTC on
September 16, 2020. Figures 1c–d display the time series of domain averaged PM$_{2.5}$ values and station counts from
the AirNow and PurpleAir networks, including matched stations between the two. PurpleAir sensors are especially
concentrated in densely populated areas, leading to notable spatial variability in observation coverage during the
September 2020 wildfire events. Coverage is particularly dense in urban regions of the western United States (e.g.
California, Oregon, Washington, Utah, Arizona and Colorado), while rural and remote areas have significantly fewer
sensors, for example, Nevada and North Dakota. The number of AirNow stations ranges from approximately 800 to
900, while PurpleAir stations number between 1,160 and 1,300. Dropouts in the AirNow network lead to sudden
decreases in station count and corresponding drops in the PM$_{2.5}$ time series. In contrast, the PurpleAir network
shows a general upward trend in station count, with no major data dropouts observed.

(a)

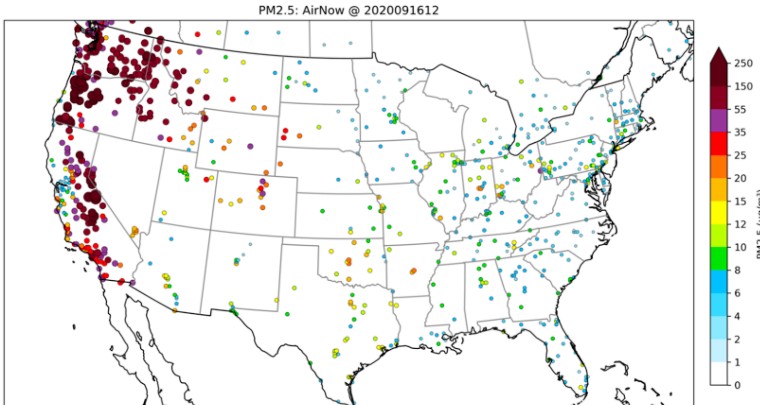










**(b)**

**(c)**

**(d)**

**Figure 1.** (a-b). Spatial distribution of AirNow(AN) and PurpleAir(PA) PM$_{2.5}$ monitoring stations on 1200 UTC 16 September 2020. (c). Time series of domain averaged PM$_{2.5}$ values and numbers from AirNow and PurpleAir observing networks. (d) Time series of domain averaged PM$_{2.5}$ values and numbers for matched AirNow and PurpleAir stations.



### 3.3 Background error covariance

In this study, the background error standard deviation ($\Sigma$) for each control variable is constructed based on the background forecast; specifically, the error standard deviations of an aerosol variable are prescribed as proportional to its background values.

The proportional scaling factor $s$ is approximately estimated by building a linear relationship between the $PM_{2.5}$ standard error ($\Sigma$) and the background forecast $PM_{2.5}^{bkg}$ $PM_{2.5}$ concentrations:

$$\Sigma = s.PM_{2.5}^{bkg} \tag{8}$$

The scaling factor $s$ is subsequently applied to all $PM_{2.5}$ components, i.e., the 70 prognostic aerosol variables, to construct their error standard deviations.

This proportionality-based approach has also been adopted in the MOCAGE operational system (Colette et al., 2024), where background error standard deviations are similarly prescribed relative to background concentrations as a first-order approximation.

Tang et al. (2023) tested a similar method, in which the background $PM_{2.5}$ error variance ($\Sigma^2$) is first estimated using the Hollingsworth–Lönnberg method (Hollingsworth and Lönnberg, 1986). A linear relationship is then established between the estimated $PM_{2.5}$ standard error ($\Sigma$) and the background forecast $PM_{2.5}^{bkg}$.

Here we take the same idea but using an alternative approach to roughly estimate the background $PM_{2.5}$ forecast error variance ($\Sigma^2$). The background $PM_{2.5}$ error variance ($\Sigma^2$) is estimated using $PM_{2.5}$ innovation information $d_b^o$ and observation error information $\varepsilon^o$ defined in the subsection 2.2, specifically,

$$E(\varepsilon^b \varepsilon^{bT}) = E(d_b^o d_b^{oT}) - E(\varepsilon^o \varepsilon^{oT}) \tag{9}$$

Equation 9 is valid under the assumption that observation and background errors are uncorrelated. This assumption is reasonable when the innovation vector $d_b^o$ is calculated using forecasts from a free-running model without any aerosol data assimilation.

In this study, short-term (e.g., 3-hour) $PM_{2.5}$ forecasts from a free run conducted during 1–21 September 2020 were used to compute the innovation vector $\boldsymbol{d}$ defined in Equation 2. This free run, referred to as the *control run*, is described in detail in the following section. Based on the innovations and observation errors defined in subsection 3.2.3, which serve as inputs to Equation 9 for estimating background standard deviation error of $PM_{2.5}$, then a scaling factor $\boldsymbol{s}$ was estimated using Equation 8, with the background $PM_{2.5}$ standard deviation error and background values as inputs. This scaling factor was subsequently applied in all assimilation experiments presented



in this study.
This proportionality-based approach implicitly assumes that displacement errors in background don't dominate,
focusing the assimilation process on correcting amplitude. It offers several benefits:
●  It helps constrain analysis increments to physically meaningful regions. For example, it prevents the

generation of sea salt aerosol increments over inland areas where no sea salt is present in the background.

This is a problem that can occur when using GSI's height-dependent or latitude–height-dependent

background error variance formulations, particularly when individual aerosol species are used as control

variables.

●  It introduces location- and time-dependent background error variance information, improving the realism of

background error specification. Moreover, the aerosol variables that dominate background errors vary by

location and assimilation cycle, rather than being consistently dominated by the same species when using

constant static background error statistics. For example, organic and black carbon typically exhibit the

largest errors in wildfire regions and downwind areas affected by smoke, whereas other regions may be

dominated by non-organic aerosols.

An example of background error standard deviation in $PM_{2.5}$ space from a data assimilation run that assimilated
both AirNow and PurpleAir $PM_{2.5}$ is shown in Figure 2. This figure is intended to illustrate the main difference to
static constant background errors, though the actual errors used in the data assimilation experiments are the errors of
the individual aerosol control variables. It is obvious that this approach produces dynamically location- and
time-dependent varying error estimates that yields particularly large error variances during the peak fire events from
10 to 20 September 2020.

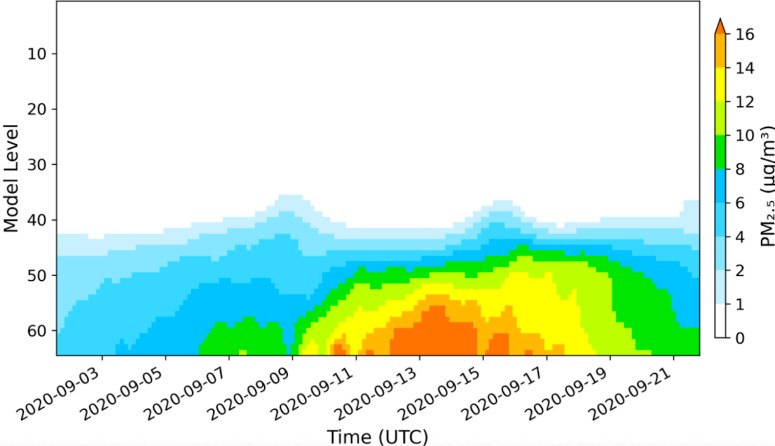


**Figure 2.** Domain averaged PM2.5 standard deviations for the data assimilation run that assimilated both AirNow

and PurpleAir PM2.5.



The background error correlation matrix C is modeled using a generic diffusion correlation operator designed for
short length scales, as implemented in the System-Agnostic Background Error Representation (SABER) repository
(Sluka, 2024). A horizontal cutoff scale of 100 km is applied, consistent with estimates derived from NMC statistics
in previous GSI applications (Wang et al., 2021). For vertical correlations, this study uses a cutoff scale of 12 model
levels, which helps confine the influence of surface $PM_{2.5}$ observations within the average daytime planetary
boundary layer (PBL) height (~1450 m) and has demonstrated improved surface $PM_{2.5}$ prediction as will be
discussed in Section 4.
**3.4 Update of total particle number and surface area concentrations**
After the aerosol mass concentration has been analyzed, total particle number concentration, total surface area
concentration can be updated accordingly. For simplicity, it is assumed that the ratio of the particle number
concentration to total particulate volume within each mode (I, J, K) remains the same as in the background. Total
particulate volume is used instead of mass mixing ratio because it is proportional to the particle number
concentration (see Eq. 3 in Binkowski and Roselle, 2003). A similar assumption was adopted by Li (2013) to update
number concentrations for the WRF-Chem model.
The number of particles is updated using the following relation:
$N_a = N_b / V_b \times V_a$   (10)
Where:
• $N_a$ and $N_b$ are the number of particles in the analysis and background, respectively, within each mode.
• $V_a$ and $V_b$ are the total particulate volumes in the analysis and background, respectively, within the same
mode.
The total particulate volume ($V_a$ or $V_b$) within each mode is calculated by dividing the mass concentration of each
aerosol variable by its corresponding density in that mode, and then summing the results. This updating approach
implicitly assumes that changes in volume across the three modes are driven solely by variations in particle number,
rather than shifts in the aerosol size distribution. The total particulate surface area within each mode is then updated
using the same volume ratio, i.e., $V_a / V_b$ (Eq.10) multiplied by the background surface area.
In preparatory work for this study, six-hourly cycling experiments (Wang et al., 2025) have shown that updating
these variables is crucial for improving AQMv7 performance. In contrast, previous work using GSI with earlier
developmental versions of AQM did not update these variables, primarily because those model versions were less
advanced than the current operational AQMv7. As a result, there was still significant room for improving prediction





skills.

**3.5 Experiments**

Table 1 provides a description of the experiments. Four experiments were conducted to evaluate the performance of
JEDI/AQM PM$_{2.5}$ DA. The first experiment is a control run (CTR), in which meteorological initial and boundary
conditions are updated every 3 hours, while chemical and aerosol fields are carried over from the 3-hour forecast of
the previous cycle. The other three experiments incorporate data assimilation: DA_AN, DA_PA, and DA_ANPA,
which assimilate AirNow PM$_{2.5}$ only, PurpleAir PM$_{2.5}$ only, and both AirNow and PurpleAir PM$_{2.5}$ observations,
respectively.
Like the CTR experiment, all DA experiments are conducted as 3-hourly cycling runs, with PM$_{2.5}$ observations
assimilated every 3 hours. 24-hour forecasts are initialized four times daily at 0000 UTC, 0600 UTC, 1200 UTC,
and 1800 UTC. The experimental period spans from 1200 UTC on September 1 to 1800 UTC on September 21,

2020.

**Table 1. Descriptions of the experiments.**

| Experiment | Data Assimilation | PM$_{2.5}$ Observations Assimilated | Aerosol Fields |
|---|---|---|---|
| CTR | No | None | Carried over from previous cycle's 3-hour forecast |
| DA_AN | Yes | AirNow PM$_{2.5}$ only | Updated by Assimilation |
| DA_PA | Yes | PurpleAir PM$_{2.5}$ only | Updated by Assimilation |
| DA_ANPA | Yes | AirNow + PurpleAir PM$_{2.5}$ | Updated by Assimilation |


# 4 Results

**4.1. Results from all cycles**

This section provides an overview of the impact of DA on PM$_{2.5}$ forecasts. A total of 80 forecasts—initialized four





times daily from 0000 UTC on September 2 to 1800 UTC on September 21, 2020—are used to evaluate model
performance. AirNow PM2.5 observations are used to verify the forecast performance. Forecast errors are assessed
using bias, mean absolute error (MAE), and root mean square error (RMSE). Forecast performance is evaluated
using box plots and performance diagrams. The box-and-whisker plots illustrate the distribution, spread, and central
tendency of forecast errors, while the performance diagrams highlight forecast skill (e.g., Critical Success Index,
CSI). Time series of $PM_{2.5}$ at various forecast hours are presented to examine the temporal evolution of forecast
performance. Additionally, spatial distributions of $PM_{2.5}$ including observations, forecasts, forecast errors, and
forecast differences are analyzed to evaluate the spatial impact of data assimilation on $PM_{2.5}$ predictions. AirNow
$PM_{2.5}$ observations are used as reference to evaluate forecast skills.
Figure 3 presents the bias, mean absolute error (MAE), and root mean square error (RMSE) for the 1–24 h forecast
of domain-averaged $PM_{2.5}$. Domain averages are computed over EPA Regions 1–10, which include all states in the
mainland United States. The detailed description of EPA regions can be found on EPA webpage:
https://www.epa.gov/aboutepa/regional-and-geographic-offices#regional, last access on 11 July 2025. Overall, all
data assimilation experiments show improved forecast skill compared to the control run. The added value of
assimilating PurpleAir $PM_{2.5}$ data alongside AirNow observations is evident in the consistent MAE reduction (Fig.
3b). Its impact on RMSE is also positive, though relatively small.

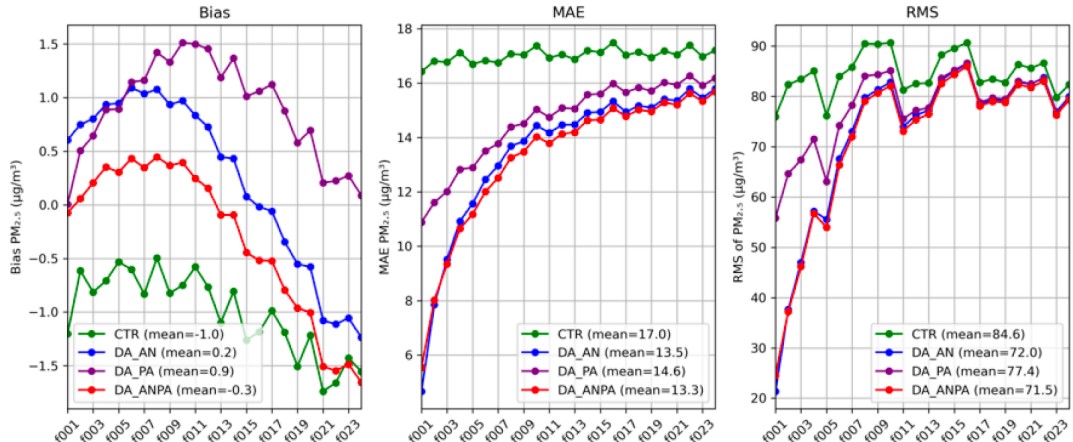


**Figure 3.** $PM_{2.5}$ forecast errors for 1–24 h lead times based on 80 forecasts initialized four times daily during
September 2–21, 2020. Domain-averaged over EPA Regions 1–10.
(a) Bias, (b) Mean Absolute Error (MAE), (c) Root Mean Square Error (RMSE).

Figure 4 shows box-and-whisker plots of $PM_{2.5}$ forecast bias. Across all forecast hours, the interquartile range
(IQR)—represented by the height of the boxes—is consistently smaller for the DA experiments compared to the
control run. This indicates reduced forecast error spread between the 25th and 75th percentiles and suggests more
consistent forecasts in the DA experiments. Although the median forecast bias in the control run is sometimes closer
to zero, the DA_ANPA experiment performs comparably in terms of central tendency while showing clear



improvements in reducing the mean forecast bias, as also reflected in Figure 3a. Among the DA experiments,
DA_AN and DA_ANPA show the most consistent improvement at 24-hour lead times, with DA_ANPA slightly
outperforming others during the early forecast hours (e.g., hour 1 to 12). This suggests that assimilating PurpleAir
observations in addition to AirNow helps reduce bias and brings the forecasts closer to observed $PM_{2.5}$ values in the
short term.

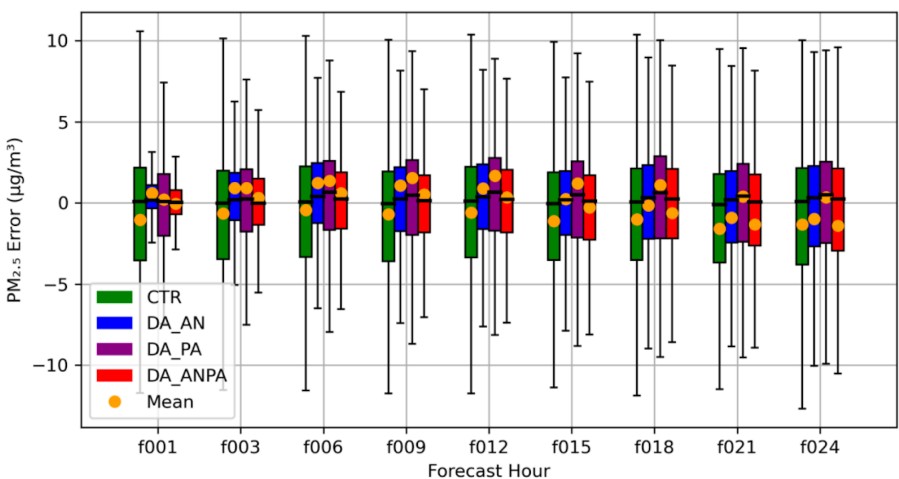


**Figure 4.** Box-and-whisker plot of $PM_{2.5}$ forecast bias. Bottom edge = Q1 (25th percentile); Top edge = Q3 (75th
percentile);Height = Interquartile Range (IQR = Q3 − Q1); Horizontal line inside box: The median (50th percentile);
Whiskers: Extend to the min and max values within 1.5 × IQR from Q1 and Q3.
Figure 5 displays performance diagrams of $PM_{2.5}$ forecast at forecast hours 1, 12, and 24 with $PM_{2.5}$ threshold of
12 µg/m³ and 35µg/m³. Performance diagrams show consistent improvements in CSI scores across all forecast hours
for all DA experiments, with DA_AN and DA_ANPA outperforming the DA_PA experiment. The performance with
$PM_{2.5}$ threshold of 5 µg/m³ (figure not shown) is similar to that of 12 µg/m³.
Figure 6 presents shows time series of $PM_{2.5}$ averaged over EPA Regions 1–10 at forecast hours 1, 12, and 24,
respectively. Consistent with the evaluations in Figures 3 and 4, all DA experiments generally improve $PM_{2.5}$
forecasts. Notably, all DA experiments help correct underpredictions during September 2–9 and 14–17. In addition,
the substantial overprediction during September 10-13 observed in the control run, largely due to inaccurate fire
emissions, is partially mitigated by the DA experiments. Among the DA configurations, DA_AN and DA_ANPA
show comparable performance and both outperform DA_PA.






(a)                                                      (b)

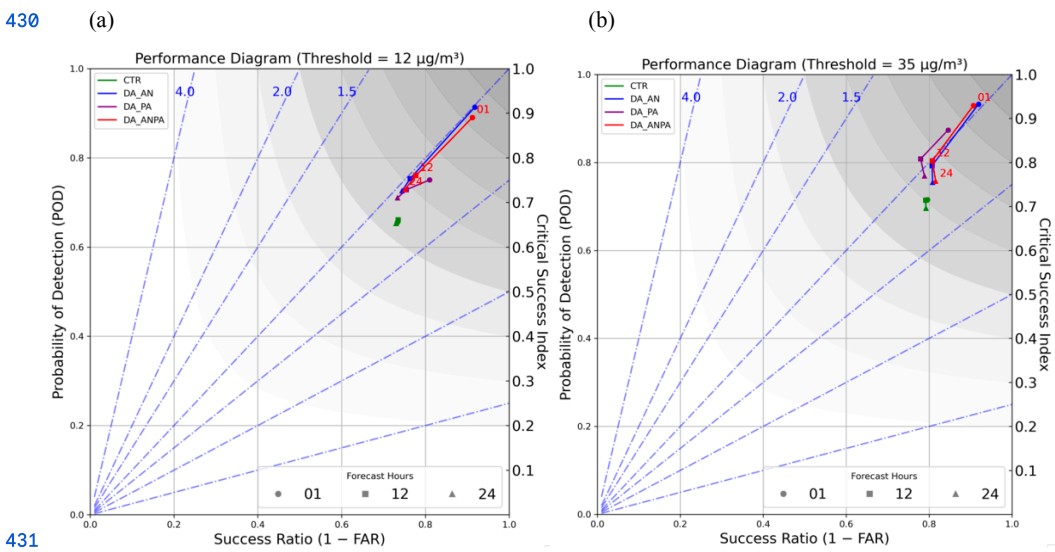

**Figure 5.** (a). Performance diagram for forecast hours 1, 12, and 24 with a $PM_{2.5}$ threshold of 12 μg/m³. (b) Same as (a), but using a threshold of 35 μg/m³.

(a)

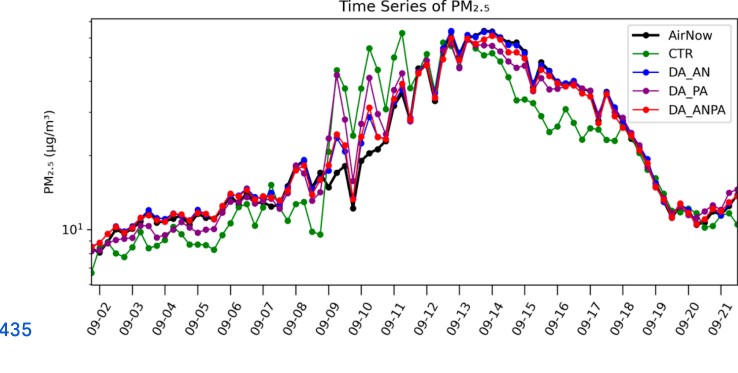

(b)

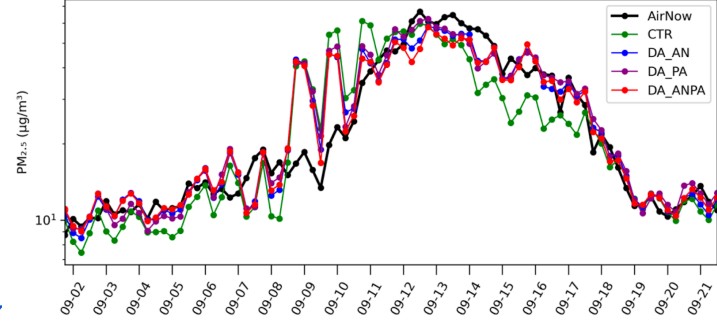



(c)

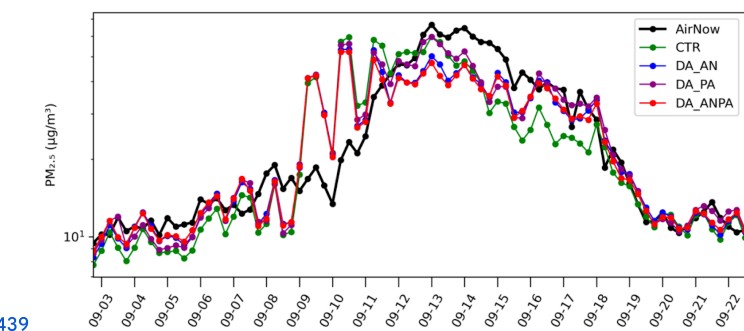

439

**Figure 6.** Time series of PM$_{2.5}$ averaged over EPA Regions 1–10 for (a) forecast hour 1, (b) forecast hour 12, and (c) forecast hour 24. The y-axis is shown on a logarithmic scale.

While we have investigated the impact of DA on PM$_{2.5}$ forecasts in terms of temporal evolution, it is also important to examine the spatial distribution of forecast fields, associated errors, and how DA influences these spatial patterns.

Figure 7 presents the spatial distribution of temporally averaged PM$_{2.5}$ forecasts at forecast hour 1, based on 80 forecasts initialized four times daily (0000 UTC, 0600 UTC, 1200 UTC, and 1800 UTC) from September 2 to 21. Figure 7a shows the PM$_{2.5}$ field from the control run (shaded), overlaid with AirNow observations. The effects of wildfire events are clearly seen across California, Oregon, and Washington—where the fires occurred—as well as in downstream regions impacted by smoke advection and transport.

Forecast errors in the control run are evident in Figure 6a but are more clearly highlighted in Figure 7b, which shows the difference between the control run and AirNow observations. Significant overpredictions appear along the California coast, as well as in parts of the Midwest and Northeast U.S., including Tennessee, Kentucky, West Virginia, and Virginia, which are approximately represented by EPA regions 1, 2, 3, 5 and 7. Conversely, notable underpredictions are found over Colorado, New Mexico, much of Texas and Oklahoma, and several Gulf Coast states which are in EPA regions 4 and 6.

Compared to the control run, both DA_AN (Fig. 7c-d) and DA_PA (Fig. 7e-f) show similar spatial correction patterns across California, Oregon, and Washington, particularly in reducing overpredictions along the California coast. They also produce comparable large-scale adjustments across the Northeast, Midwest, and Southern U.S., with error patterns (Fig. 7d and 7f) largely opposite in sign to those in the CTR–AirNow difference (Fig. 7b). This suggests that both DA experiments effectively mitigate the control run's over- and underpredictions.

However, the magnitude of correction is generally smaller in DA_PA than in DA_AN. Notably, DA_PA shows its strongest impact over Nevada, northern Utah, Colorado, and southwestern New Mexico, helping to alleviate the underpredictions in these regions—similar to improvements seen in DA_ANPA.





**463 (a) CTR**
**(b) CTR-AirNow**

**465 (c) DA_AN**
**(d) DA_AN-CTR**

**467 (e) DA_PA**
**(f) DA_PA-CTR**

**469 (g) DA_ANPA**
**(h) DA_ANPA-CTR**

**Figure 7.** Spatial distribution of average PM$_{2.5}$ at forecast hour 1, based on 80 forecasts initialized four times daily
(0000 UTC, 0600 UTC, 1200 UTC, and 1800 UTC) during 2–21 September.
(a) PM$_{2.5}$ in experiment CTR (shaded) overlaid with AirNow PM$_{2.5}$ observations (filled dots).



(b) PM$_{2.5}$ bias in experiment CTR.
(c) PM$_{2.5}$ in experiment DA_AN (shaded) overlaid with AirNow PM$_{2.5}$ observations.
(d) PM$_{2.5}$ difference between experiments DA_AN and CTR.
(e) PM$_{2.5}$ in experiment DA_PA (shaded) overlaid with AirNow PM$_{2.5}$ observations.
(f) PM$_{2.5}$ difference between experiments DA_PA and CTR.
(g) PM$_{2.5}$ in experiment DA_ANPA (shaded) overlaid with AirNow PM$_{2.5}$ observations.
(h) PM$_{2.5}$ difference between experiments DA_ANPA and CTR.

Figure 8 shows the percentage change in MAE (%) between the DA experiments and the control (CTR) experiment
at forecast hours 1 and 24. Negative values indicate a reduction in MAE. All DA experiments show improvements:
at forecast hour 1, MAE is reduced by approximately 60% in DA_AN and DA_ANPA, and by around 18% in
DA_PA. At forecast hour 24, reductions are smaller but still present—about 6–7% in DA_ANPA and around 2% in
DA_PA. MAE is reduced at nearly all stations at forecast hour 1, and at most stations by forecast hour 24.
In summary, DA also improves the spatial distribution of PM$_{2.5}$ forecasts. Figure 7 shows that the control run
exhibits significant regional biases, with overpredictions along the California coast and in parts of the Midwest and
Northeast (EPA regions 1, 2, 3, 4, and 7), and underpredictions in the Southwest and Gulf Coast (EPA regions 4 and
6). All three DA experiments reduce these errors, particularly correcting coastal overpredictions and improving
forecasts in regions affected by wildfires. While DA_PA provides slightly weaker corrections than DA_AN, it
contributes meaningful improvements in the Mountain West and Southwest.












DA_AN vs. CTR

DA_PA vs. CTR

DA_ANPA vs. CTR

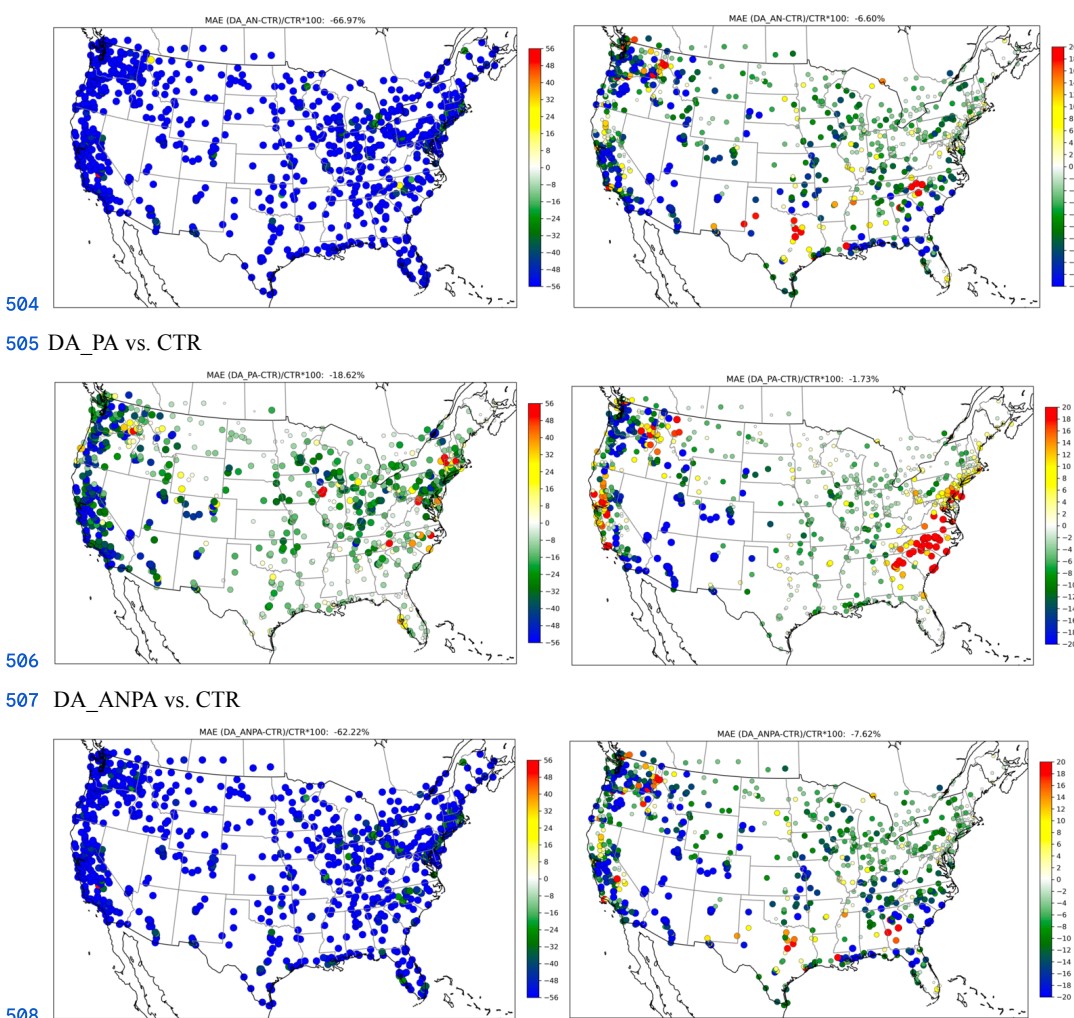

**Figure 8.** Percentage change in MAE (%) between DA experiments and the control (CTR) experiment. The percentage is calculated as (MAE(DA)-MAE(CTR))/MAE(CTR)*100.

Left panels show the 1-hour forecast; the right panels show the 24-hour forecast.

Top row: DA_AN vs. CTR; Middle row: DA_PA vs. CTR; Bottom row: DA_ANPA vs. CTR.

## 4.2. Results from Forecasts Initialized at 1200 UTC

In this section, we examine the forecasts initialized at 1200 UTC, which is the time when the operational AQM launched a 72-hour forecast. Time series of 1–24-hour $PM_{2.5}$ concentrations are analyzed by grouping the EPA regions into three areas:



- Area A includes EPA Regions 8, 9, and 10, which are the areas where fires occurred and/or were most influenced by smoke.
- Area B includes EPA Regions 1, 2, 3, 5, and 7, where the control run generally shows an overall overprediction (see Fig. 6b).
- Area C includes EPA Regions 4 and 6, which show an overall underprediction in the control run (also shown in Fig. 6b).

Time series of 1–24 h $PM_{2.5}$ forecasts for the above three areas are shown in figure 9. There is a large overprediction (spikes) in Area A from September 9 to 11 in the control run, followed by a transition to underpredictions from September 12 to 17. The overprediction is due to inaccurate fire emissions, as similar spikes are not observed in Area B and C. However, this overprediction contributes to spurious "good" performance in the control run during the transition period from September 11 to 12. As shown during this period, the later (~3–24 h) forecasts from the DA_AN and DA_ANPA experiments do not outperform the control run, although their first-hour forecasts are closer to AirNow observations. Overall, the DA experiments clearly improve both the overprediction from September 9 to 11 and the underprediction from September 14 to 17.

Regarding the forecasts over Areas B and C, the overprediction in Area B and the underprediction in Area C are generally improved.

Figure 10 shows $PM_{2.5}$ forecast error statistics (Bias, MAE, and RMSE) for forecast hours 1–24. At the first forecast hour, the DA experiments in Areas A, B, and C outperform the control run across all three metrics. In terms of MAE and RMSE, DA_AN and DA_ANPA perform better than both DA_PA and the control run in Areas A and B. For the 1–24 h $PM_{2.5}$ forecasts in Area C, all DA experiments outperform the control run in terms of Bias and MAE. However, RMSE improvements are only seen up to forecast hours 7–9.

The impact of additional assimilation of PurpleAir data shows area-dependent behavior. For example, it slightly reduces MAE and RMSE in Area A and noticeably reduces Bias, MAE, and RMSE in Area B. In Area C, no clear positive impact is observed, although assimilating PurpleAir data alone still results in better performance than the control run.

Performance diagrams (Fig. 11) show consistent improvements in Critical Success Index (CSI) scores across all forecast hours for all DA experiments compared to the control run. Among the DA configurations, DA_AN and DA_ANPA show comparable performance and both outperform DA_PA at 1h and 12h forecasts.





**(a)**

**(b)**

**(c)**

**Figure 9.** Time series of 1–24 h PM$_{2.5}$ forecasts: (a) averaged over EPA Regions 8–10; (b) averaged over EPA Regions 1, 2, 3, 5, and 7; and (c) averaged over EPA Regions 4 and 6.









**Figure 10.** PM$_{2.5}$ forecast statistics at forecast hours 1–24. Top row: averaged over EPA Regions 8–10.; Middle
row: over EPA Regions 1,2,3,5,7, and Bottom row: over EPA Regions 4,6.




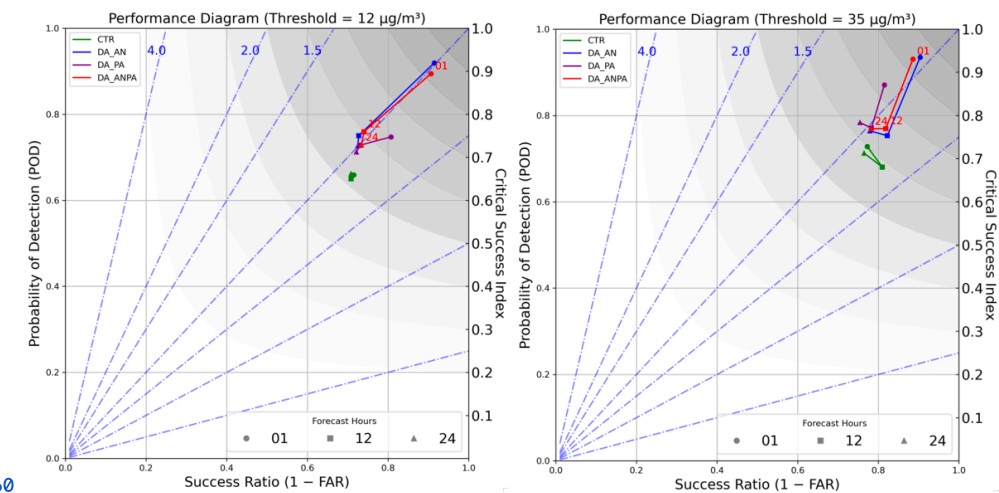


**Figure 11.** Performance diagram for forecast hours 1, 12, and 24 with a PM$_{2.5}$ threshold of (a) 12 µg/m³, (b)
35 µg/m³.

## 5 Summary and discussion

The latest version of NOAA's regional AQM system was implemented and became operational on May 14, 2024.
This system has been developed through the online coupling of the Finite Volume Cubed Sphere (FV3) atmospheric
model with the Environmental Protection Agency (EPA)'s Community Multiscale Air Quality (CMAQ) model
within the Unified Forecast System (UFS) framework. In order to provide improved initial conditions for AQM
supporting enhanced prediction of wildfire impacts on air quality prediction, the capability to assimilate PM2.5
observations into AQMv7 was developed within JEDI and tested using its 3D-Var assimilation component. Note that
the developed assimilation scheme can also be used to generate analysis (reanalysis) dataset for other applications,
for example, providing data for training artificial intelligent models used in air quality prediction.

Data assimilation experiments were conducted for the September 2020 Western U.S. wildfire episode, using
3-hourly cycling with observations from the AirNow and PurpleAir networks. Results showed that assimilating
AirNow PM$_{2.5}$ observations significantly improved 1–24 hour forecast skill. Mean absolute error (MAE) was
reduced by approximately 60% at forecast hour 1 and 7% at hour 24, relative to forecasts without data assimilation.
Assimilating PurpleAir data alone yielded more modest improvements—approximately 18% at hour 1 and 2% at
hour 24—but when combined with AirNow, PurpleAir data provided additional benefit by further reducing MAE
slightly either compared with AirNow observations (Fig. 3) or PurpleAir observations (Figure not shown). The
positive impact of the PurpleAir data assimilation during the September 2025 wildfires was also demonstrated in an
experimental Rapid Refresh Forecast System coupled with Smoke and Dust Model (Wang et al., 2023), where it
significantly reduced the model's 24-h underprediction of surface PM$_{2.5}$. Considering the PurpleAir data coverage





has improved since September 2025, the results of this study further highlight its potential to complement AirNow
observations by filling spatial gaps and improving PM$_{2.5}$ forecast skills.
In this first implementation of aerosol data assimilation in JEDI for AQMv7, the control variables are defined as
individual forecast aerosol variables. In previous work on aerosol data assimilation for an earlier version of AQM
using the GSI system (Wang et al., 2021), one option for the control variables was to define them as the total aerosol
mass in each of the three modes, resulting in just three control variables. A control variable transform (CVT) was
then applied to partition the analysis increments across these modes to individual aerosol species, based on the ratio
of each species' mass to the total mass within the corresponding mode. The use of total aerosol mass in the three
modes as control variables—thereby reducing the number of control variables from 70 to 3—is planned for a future
phase of development. The use of total masses as control variables also reduces the cost of the background error
statistics calculation and iterative minimization. (Kumar et al. 2019). It is noted that the ensemble based data
assimilation approach is superior to capture flow-dependent background error covariances and aerosol assimilation
along with emission updates can be developed when an ensemble prediction system for AQM is there.
This study focused on surface-level PM$_{2.5}$ and did not incorporate vertical profile constraints with satellite-based
aerosol optical depth (AOD) retrievals, which could further enhance forecast skill. A key challenge is the need for a
robust forward operator in the CRTM AOD module—specifically, the creation and validation of lookup tables
(LUTs) for AOD calculations with AQM. As an intermediate solution, existing LUTs in CRTM, such as the GEOS-5
LUTs, have been tested by grouping and mapping AQM aerosol species to those used in GEOS-5 (Wang et al.
2025). However, this approach presents several issues. For instance, AQM does not distinguish between hydrophilic
and hydrophobic aerosol types of organic carbon and black carbon, whereas GEOS-5 does. Additionally, AQM
(through CMAQ) uses a modal aerosol representation, while GEOS-5 adopts a bin-based approach, making the
mapping between the two systems non-trivial. AOD assimilation also depends on an accurate vertical distribution of
aerosols in the background field so that the CRTM AOD operator can provide meaningful gradient information at
the correct vertical levels to constrain the analysis update. However, AQM models have shown deficiencies for the
September 2020 fire events in representing smoke concentrations at and above plume rise levels, largely due to how
fire emissions are injected into the model. This will be improved in the next update of the operational AQM.

## 609 Code and data availability

The AQMV7 model, JEDI software and PM$_{2.5}$ and fire emission data we used in this research are publicly available
on on Zenodo (https://doi.org/10.5281/zenodo.17049857; Wang et al., 2025b).
Users are referred to the guidance on compiling and running the model:
https://ufs-srweather-app.readthedocs.io/en/develop/UsersGuide/index.html (Last accessed on August 26, 2025).
Global Forecast System analysis data were downloaded from the NCAR Research Data Archive:
https://doi.org/10.5065/D65D8PWK (last access: Aug 26 2025)



## Author contribution

HW designed and developed the $PM_{2.5}$ DA capability within JEDI for the AQM model, conducted experiments, and evaluated performance; CM and JB contributed to $PM_{2.5}$ DA methodology, advised on code implementation, and assisted in performance analysis; SW contributed to $PM_{2.5}$ DA methodology and experimental design; RL conducted control experiments and contributed to workflow development; JL and KW contribute to model configuration and control run setup; YT contributed to background error modeling and observational error specification; HC, AT and HL contributed to workflow development; JL performed quality control and correction of PurpleAir observations.

## Competing interests

The authors declare that they have no conflict of interest.

## Acknowledgements

Thanks Dr. Mohmmed Farooqui at Texas A&M University-Kingsville for assisting in Python scripts to download the PurpleAir observations.

This research was supported by  the Fire Weather and Precipitation Research and Development in Support of the Disaster Relief Supplemental Appropriations Act (DRSA) project (NA23OAR4050200D), and in part by a NOAA Cooperative Agreement NA22OAR4320151 with the University of Colorado. The scientific results and conclusions, as well as any views or opinions expressed herein, are those of the authors and do not necessarily reflect those of NOAA or the Department of Commerce.

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
