# Peer review of "PM2.5 Assimilation within JEDI for NOAA's Regional Air"

_EGUsphere, 2025_

## Referee Comment (RC1)

**Comments to the Authors**

**General comments:**

The study by Wang et al. provides a PM2.5 data assimilation technique in a regional air quality model (AQMv7) and apply the modelling system to September 2020 western US wildfires. The authors conducted several experiments including a control run without any assimilation and runs with AirNow and PurpleAir data assimilated to demonstrate the improved forecast performance with data assimilation. The manuscript fits in the scope of the journal. However, there are several unaddressed issues that limits the strength of the manuscript for the reader in its current form. Hence I strongly suggest that the following comments are addressed before consideration of the manuscript for publication.

**Major comments:**

- (1) How is this study different than previous similar studies conducted over US (for example, compared to US based studies mentioned in lines 79-80). Novelty part should be articulated clearly.
- (2) Usually for studies utilizing the air quality models, physics and chemistry options selected are usually reported. Does this model use a fixed suite of options or there are several options available for physical and chemical processes. In any case, the options used should be reported somewhere. Also, how are meteorological initial and boundary conditions prepared?
- (3) There is no clear information on emission inputs? Which anthropogenic and natural emissions are included in the model.
- (4) It would help the reader if there is a flow chart to explain the complete system used in this study.
- (5) Lines 245-246 Why are the error standard deviations different for AirNow and PupleAir and how the results would change if error standard deviations are kept same for both AirNow and PupleAir (to say 10%). In other words, would it make more sense to compare DA PA and DA AN if error standard deviations are kept same?
- (6) Line 255-256 Explain more clearly about these different thresholds for large PM2.5 observations mentioned for AirNow and PurpleAir?
- (7) Equation 9 is difficult to understand. Also, what is T in superscript?
- (8) Is there any possible explanation for the hourly changes (for example first increase and then decrease in bias) in figure 3 and other similar figures like figure 10. Also, put axis title on the x-axis in figure 3?
- (9) Line 418-421 Can you briefly explain about performance diagram with some references? How are these thresholds chosen?
- (10) Line 455-459 and figure 7 I think it will be better if bias is shown for DA\_AN, DA\_PA and DA\_ANPA with respect to AirNow (as a reference set of observations) for ease of comparison between figures 7 (b,d,f,h) for the forecast performance.

- (11) Line 482-486 Can the authors also discuss the performance EPA region-wise (as in lines 517-522) for 1h and 24h forecast. This may help to understand the performance in wildfire impacted regions more clearly.
- (12) Lines 531-532 Mention some statistics to demonstrate improvement.

**Minor comments:**

- (1) Line 22 particles with aerodynamic diameters
- (2) Line  $40-41 \text{"PM}_{2.5}$  is a major contributor to poor air quality in US" Provide some references to this line.
- (3) Line 44 Kindly avoid the usage of term 'primary pollutants' in this context.
- (4) Line 76 replacing the previous offline-coupled the Global Forecast System
- (5) Line 77-78 "modeling system" repeated twice.
- (6) Line 88 Put 2.5 here (and in other places in manuscript) as subscript
- (7) Line 170 Check the sentence. Should it be: *It is noted that a recipe that uses temperature...*
- (8) Line 203 Mmountain Rregion?
- (9) Line 213 Add the last access date.
- (10) Line 242 and Texas.
- (11) Lines 309-312 Reduce the sentence length. Break into two sentences for ease of read.
- (12) Lines 345-346 Is it "the ratio of the particle number concentration to total particulate volume"? Particle number concentration is already number per unit volume.
- (13) Line 422 Figure 6 presents shows time series of
- (14) Line 580 September 2025 or 2020?
- (15) Line 593 There are two full stops before and after the reference.

---

## Referee Comment (RC3)

The manuscript titled " PM2.5 Assimilation within JEDI for NOAA's Regional Air Quality Model (AQMv7): Application to the September 2020 Western U.S. Wildfires" by Wang et al. with reference egusphere-2025-4098 is a valuable scientific contribution in operational aerosol assimilation, particularly for wildfire smoke events. Overall, it is very well written and provides a detailed investigation of the PM2.5 assimilation of AirNow and PurpleAir surface observations in AQMv7 for the US in September 2020. The spatio-temporal investigation of the influence on different forest lead times (1-24hours) provides interesting insights.

Overall, I suggest accepting the manuscript with the following mayor corrections:

**General comments:**

A) My first general comment is on the specific purpose of this paper: For me, it is not clear wether the novelty of the contribution is about (1) the general implementation of an PM2.5 assimilation system for surface stations, or (2) just about the impact of additionally assimilating PurpleAir observations. While the text (title, abstract, introduction) suggest (1), some parts remain confusing in this regard:
  1. In l. 81-84 you mention other literature assimilating AirNow PM2.5 in the US. Please specify the distinctive novelty of this contribution compared to previous work.
  2. It is not clear in Sec.2 what the new developments are and what was already available before. Please state more clearly in the beginning of the section or for each subsection.
     - incl l.129: is this part of the new developments?, otherwise please add reference
     - incl l.159ff: (same)

  3. Along these lines, it is also not clear in the summary in Sec.5 if l. 565-572 is about new developments. Please specifically summarize the purpose and novel developments of this study in Sec.5 (see also general comment D).

B) Sec. 4.1 contains plenty of plots which are only discussed shortly. This is fine, but I would suggest moving some less important plots in the Apx. (e.g. Fig.5, Fig.8) to increase readability. Or divide into subsections and consider expanding the description of results from each plot (eg temporal, spatial comparison, especially if Sec.4.2 is deleted/shortened: see general comment C)

C) Sec. 4.2: Concerning the whole content of Sec.4.2: The only new result I can see in this section is l.538-541. All the rest seems to reproduce the results of Sec.4.1. I suggest removing this whole section, and maybe putting some of the plots in the Apx. (eg Fig.10 if the results in l.538-541 are considered important). Otherwise:
  ○ Please explain more clearly why you are specifically looking at the forecast initialized at 12UTC. You mention the operational 72-hour forecast, but only show results for forecast hour 1-24. What additional information does this investigation provide compared to the overall results above?
  ○ e.g. Given the fact that I don't know where Areas B and C are exactly (see smaller comment 15): What is the additional result of l.531f compared to Fig.7? Also l.533-537, l.542-544
  ○ Please restrict this section to specifying and discussing the novel results that cannot be obtained from Sec.4.1.
D) Sec.5: I'm missing an actual summary and discussion of the results of this study. What conclusions do you draw from the results above? You show plenty of plots with different

aspects in Sec.4, I would wonder if the only important aspects you want to summarize and discuss are in l.573-584.

1. e.g. Do you have any explanation or conclusion from the temporal and spatial differences between the runs that you describe in Sec. 4?

2. The remaining parts of Sec.5 are about previous work (as far as I understand, l.565-572, compare general comment A:3) and outlook (l.585-608).

**Smaller comments:**
1. Sec.3 includes lots of different subsections. I would suggest giving a short overview of the content at the beginning of the Sec.3.
2. Sec.3.1: Please provide some more general information on the region that you're discussing, eg west/east? which state?... For a non-US reader it's not obvious where the Willamette Valley is. And please specify what you are referring to as "broader region"?
3. l. 234f: Which reference PM2.5 data did you use in your modified regression equation? Please specify.
4. l. 244-256: Please restructure this paragraph. If I understand correctly, you're switching between AirNow (l.245, l.247f, l.250ff) and PurpleAir (l.246, l.248f) multiple times. Please formulate more clearly.
   - l. 247f: Does this refer to PurpleAir? Please specify.
   - l. 248f: Redundant information with l.245.
   - l.249 "5%": For comparison to your setup, please specify the type of observations they assigned with 5% errors, eg lowcost or "AirNow-like" stations?
   - l. 253: Unit of "1.5" missing (should be the same as PM2.5 concentrations, no?)
   - l. 254ff: Unclear, please specify where these numbers coming from. Is this a result from the spatial averaging to a 0.1deg grid? As far I understand, this can only reduce the effect of large PM2.5 observations if there are always multiple observations within one gridcell. Is this always the case? Please explain.
5. l. 259: Please specify the matched stations in Fig.1d. Does this refer to locations where both networks have closely located stations? If yes, which criteria did you use to define "closely located"?
   - At the end of this subsection, it is also not clear why you are looking at the matched values in Fig.1d. What does it tell you? Is there any consequence you are taking from it? Otherwise, remove Fig.1d.
6. l. 289: Is the linear relationship the same for all 70 variables? Please describe.
7. l.308: inconsistent notation of innovation vector: here(l.308) bold d, Eq.2: non-bold d, Eq.9: non-bold d with subscript b and superscript o. Please unify or define differences.
8. l. 338: I would associate a "cutoff scale" this to be the distance at which the correlation function is "cut off", assuming zero correlation beyond that. But this is different from a correlation length that determines the shape of the correlation function itself. Please specify.
9. l. 374: Please specify what happens to the PM2.5 field at the initialization times. Do you initialize from the last analysis? Since you assimilation PM2.5 every 3 hours, what is the difference in PM2.5 between the assimilation times and the initialization times? Or does the initialization only apply to meteorological fields?
10. l. 383: Are these independent observations left out for the experiments that assimilate AirNow? And if not: How can the bias of DA_AN be worse than DA_ANPA (and DA_PA) during the first forecast hours, if compared agianst the same assimilated AirNow data (Fig.3)? Please discuss the implications of validating the different experiments with AirNow data concerning potentially temporally-correlated errors.
11. Fig.3: Please discuss why the bias of the joint assimilation DA_ANPA is closer to CTR than each of the single-obs. assimilations DA_AN and DA_PA. Please also discuss the change in

biases over time, i.e. increasing underestimation with forecast time which makes the single-obs. assimilations DA_AN,DA_PA being least biased for longer forecast hours.

12. l. 458f: This can hardly be quantified from Fig7. For the assimilation runs, it would be helpful to show their differences to AirNow validation observations. I would suggest e.g. removing their absolute PM2.5 fields (Fig.7c,e,g) and showing their differences to AirNow validation along with their differences to the control (keeping Fig.7,d,f,h).

13. l. 483ff: I assume you refer to mean reductions (domain-averaged)? Local reductions seem to be much larger and varying. Moreover, when you only discuss mean reductions, this does not fit to in the discussion on spatial distributions here (introduced in l.442f, summarized in l.487ff). Please either discuss spatial distributions OR
    ○ show only mean reductions in the figure (replacing Fig.8)
    ○ and move this paragraph where it fits

14. l. 491f: This was not mentioned above. From which plot do you see this? Please specify.

15. l. 517-522: Please write a bit more where these Areas A,B,C are located (eg states). For external readers, it's hard to guess just from the given information.

16. l. 525f: Is this a guess? Or how do you conclude this?

17. l. 579-582: The sentence suggests that these results from the given reference. But the reference is from 2023 while the case study is from 2025, that's not possible. Please clarify.

18. l. 582f: Now it's November 2025, I assume the data coverage did not change significantly within 2 months. Furthermore, you only use data from 2020 in this study. Is there maybe a mistake in the year (see also comment above)? Or I completely misunderstand this paragraph.

19. l. 585-595: Confusing. You talked about the results before (l. 573-584), and now afterwards about the implementation (l.585-595). I suggest restructuring.

20. l. 596-608: I'm not sure if you need to discuss these challenges in that much detail here. This work is about assimilation of surface observations, and in my point of view it's enough here to say that you are planning to include improved AOD assimilation into the system.

**Technical and formulation-related comments:**
- l. 27-29: That's a very technical sentence for the abstract. At this point, it is not clear what control variables are in this context. And how background error standard deviations scale to background state values. It looks like quite a lot of information was squeezed into one sentence. Is it necessary to be included in the abstract? If yes, please expand, otherwise I suggest removing here.
- l. 34: please explain the abbreviation "CONUS" once.
- l. 45f: Do you refer to the specific AQI of EPA, or in general to any AQI? Please specify.
- l. 179-183: Is there a square missing somewhere? The diagonal of a covariance is the variance, squared standard deviation. So Sigma should be the variance matrix, or it has to be Sigma^2 in Eq.(5). (also in l.184ff).
- l. 198f: please specify. In that region (which region, see smaller comment 2)? Or in whole US?
- l. 240ff: Can this be seen from Fig.1b? Please add reference to plot.
- l. 257f: The two sentences introducing different plots in Fig.1a-b and Fig1.c-d are confusing, because they are disconnected from their descriptions/interpretations below. I suggest moving the sentences referring to Fig.1a-b (l.259-263) directly after mentioning Fig.1a-b here. And moving the sentence introducing Fig.1c-d down right before its description in l.263ff.
- l. 297-301: You are mentioning multiple times that background PM2.5 error variance is denoted as Sigma^2. That might be confusing. I'd suggest defining once, and using either words or symbol afterwards.

- l. 327: Technically, Fig.2 is not not PM2.5 space. This would be eg showing the background error standard deviation as function of PM2.5. Do you mean the PM2.5 background error standard deviation as (weighted) sum over all 70 variables?
- l. 389f: This sentence is doubled with l.383.
- Fig.4: Description of mean missing in figure caption.
- l. 422: two verbs ("presents shows")
- Fig.5, 7, 8, …: Icons and labels are very small and hard to see/read.
- l. 523: capital "F" in "figure"
- l. 593: remove "." before reference
- l. 593ff: is an ensemble prediction system planned for AQM? Please explain or remove sentence.

---

## Author Comment (AC1)

General comments:

The study by Wang et al. provides a PM2.5 data assimilation technique in a regional air quality model (AQMv7) and apply the modelling system to September 2020 western US wildfires. The authors conducted several experiments including a control run without any assimilation and runs with AirNow and PurpleAir data assimilated to demonstrate the improved forecast performance with data assimilation. The manuscript fits in the scope of the journal. However, there are several unaddressed issues that limits the strength of the manuscript for the reader in its current form. Hence I strongly suggest that the following comments are addressed before consideration of the manuscript for publication.

We thank the reviewer for the thorough evaluation of our manuscript and for the constructive comments and suggestions.

In response to the reviewer's comments, we have revised the manuscript accordingly and addressed the issues raised to strengthen the clarity and robustness of the study. Below, we provide point-by-point responses to the reviewer's comments, with **our replies shown in blue font. The changes in the revised manuscript are also shown in the blue font.**

Major comments:

(1) How is this study different than previous similar studies conducted over US (for example, compared to US based studies mentioned in lines 79-80). Novelty part should be articulated clearly.

Thank you for pointing this out. We have revised and expanded the manuscript to more clearly describe the novelty of this study relative to previous work. A new paragraph has been added to summarize the unique aspects of this research (see also the second-to-last paragraph of the Introduction), and the revised text is provided below.

> "This study aims to develop and evaluate an initial aerosol analysis capability for the AQMv7 system by assimilating $PM_{2.5}$ observations using the JEDI 3D-Var framework. Compared to previous $PM_{2.5}$ data assimilation studies, this research adopts the NOAA's regional operational AQMv7 system and incorporates a new $PM_{2.5}$ transform in JEDI for assimilating $PM_{2.5}$ observations. In addition to evaluating the impact of assimilating AirNow $PM_{2.5}$ measurements on air quality prediction, this study also examines the impact of assimilating low-cost PurpleAir observations. Although PurpleAir data are valuable for real-time air quality monitoring, their impact on numerical air quality prediction has not been thoroughly investigated. To the authors' best knowledge, this is the first study to demonstrate the value of PurpleAir observations for air quality prediction during a major fire event using the AQMv7 system."

(2) Usually for studies utilizing the air quality models, physics and chemistry options selected are usually reported. Does this model use a fixed suite of options or there are several options available for physical and chemical processes. In any case, the options used should be reported somewhere. Also, how are meteorological initial and boundary conditions prepared?

The model configuration of the operational NOAA regional AQMv7 system is described in detail by Huang et al. (2025). In that Bulletin of the American Meteorological Society paper (open access; DOI: 10.1175/BAMS-D-23-0053.1), the authors provide a comprehensive description of the AQMv7 configuration, including the physical and chemical parameterizations, emissions, and meteorological initial and boundary conditions.

In the revised manuscript, a new paragraph is added in Section 3.1 to summarize the physics and chemistry options, as well as the meteorological initial and boundary conditions. Here is the new paragraph:

> "In this research, the model configuration is almost the same as the operational AQMv7 setup except for running over the CONUS domain with a 3 hourly cycling interval. The AQMv7 system is configured over the CONUS domain with a grid-spacing of 13 km and 65 vertical levels, extending up to 0.2 hPa. The system uses the Global Forecast System version 16 (GFSv16) physics package within the Common Community Physics Package (CCPP) framework to generate the meteorological fields driving air quality predictions. Meteorological initial conditions and lateral boundary conditions are generated using GFS forecast outputs with lead times up to 30 hours at 3-hour intervals from the previous GFS cycle. Fire-related emissions are represented using real-time Regional hourly Advanced Baseline Imager (ABI) and Visible Infrared Imaging Radiometer Suite (VIIRS) Emissions (RAVE) data at 0.03° spatial resolution. Anthropogenic emissions are based on the 2016 U.S. EPA NEI Collaborative (NEIC2016v1) modeling platform. Gas-phase chemistry is simulated using the Carbon Bond Mechanism version 6 (CB6r3) with updated isoprene chemistry and revised photolysis rates. More detailed information including physics, chemistry options, anthropogenic emissions, and fire emissions about the model configuration can be found in Huang et al. (2025). "

Readers are referred to Huang et al. (2025) for a complete description of the model setup.

(3) There is no clear **information on emission inputs**? Which anthropogenic and natural emissions are included in the model.

Please see the reply to the previous comment. The emission info can be found in a new paragraph in Section 3.1.

(4) It would help the reader if there is a flow chart to explain the complete system used in this study.
A schematic of the data assimilation and forecasting cycles was added into Section 3.5 Experiments.

(5) Lines 245-246 - Why are the error standard deviations different for AirNow and PupleAir and **how the results would change if error standard deviations** are kept same for both AirNow and PupleAir (to say 10%). In other words, would it make more sense to compare DA_PA and DA_AN if error standard deviations are kept same?

The quality of AirNow data is superior to the low-cost PurpleAir PM data.

Below are reasons that PurpleAir PM was assigned larger errors than AirNow PM. PurpleAir sensors are designed for affordable, wide deployment by individuals or community groups. They use low-cost optical particle counters to count particles and estimate PM2.5 mass concentrations. These data require

calibration to improve accuracy because the sensors measure particle counts and infer mass concentration using assumptions that do not always match real-world aerosol properties. In addition, the sensors are installed and maintained by individuals, either outdoors or indoors. For our application, PurpleAir data are reported by outdoor sensors, and must undergo quality control (QC) and apply appropriate corrections before being assimilated.

In contrast, AirNow/EPA data are collected using regulatory-grade instruments that physically collect and weigh particles. These instruments are installed, maintained, calibrated, and quality-checked by trained agencies, ensuring high accuracy and comparability. As a result, no additional QC or correction is required prior to data assimilation. The current observational errors setup might not be an optional one, but it is reasonable to initiate data assimilation cycles and evaluate data impact.

There are other objective methods to estimating observation errors in a data assimilation system. This will be investigated in the near future.

(6) Line 255-256 - Explain more clearly about these different thresholds for large PM2.5 observations mentioned for AirNow and PurpleAir?
The above lines were removed in the updated manuscript.

(7) Equation 9 is difficult to understand. Also, what is T in superscript?
In Equation (9), the superscript $T$ denotes the transpose of a vector, and $E(\cdot)$ denotes the mathematical expectation operator. Equation 9 has been revised and its description extended in the revised manuscript. The notation has also been updated to ensure consistency with that used in the previous sections.

(8) Is there any possible explanation for the hourly changes (for example - first increase and then decrease in bias) in figure 3 and other similar figures like figure 10. Also, put axis title on the x-axis in figure 3?

We did discuss the temporal bias evolution and explained the bias at forecasts 1 from control and data assimilation experiments in the revised manuscript.

In general, the data assimilation experiments exhibit a similar temporal trend to the corresponding forecasts without data assimilation as data assimilation primarily corrects the model state and does not resolve inherent model bias.

This paper primarily focuses on the development of the data assimilation scheme and the evaluation of data impacts. We assume that the observed temporal bias evolution is mainly controlled by diurnal cycles/variations but dominated by certain forecast cycles and their interactions between fire emissions and meteorological processes, particularly boundary layer physics (e.g., the diurnal evolution of PBL height).

We have also added an explanation of the x-axis in the caption of Figure 4 as requested. Specifically, the x-axis represents forecast lead times from 1 to 24 hours.

(9) Line 418-421 - Can you briefly explain about performance diagram with some references? How are these thresholds chosen?

We have restructured this section to focus more closely on the main results and conclusions of the study. As part of this revision, the two figures showing the performance diagrams have been removed from the revised manuscript. Consequently, a detailed explanation of the performance diagram and the associated threshold selection is no longer included.

(10) Line 455-459 and figure 7 – I think it will be better if bias is shown for DA_AN, DA_PA and DA_ANPA with respect to AirNow (as a reference set of observations) for ease of comparison between figures 7 (b,d,f,h) for the forecast performance.
AirNow observations are already overlaid on the 1-h forecasts from each experiment in Figures 7a, c, and e. The purpose of these panels is to illustrate how the 1-h forecasts from the data assimilation experiments are improved relative to the control (CTR) run. For this reason, the CTR experiment is used as the reference in Figures 7d and f, which highlight the relative impact of data assimilation. By comparing these panels with Figure 7b, the improvements in the 1-h forecasts from the data assimilation experiments can be more clearly identified.

(11) Line 482-486 – Can the authors also discuss the performance EPA region-wise (as in lines 517-522) for 1h and 24h forecast. This may help to understand the performance in wildfires impacted regions more clearly.
We appreciate the reviewer's suggestion.
Following another reviewer's recommendation, we restructured the results section. A figure shows the statistical performance metrics for averaged 1–24 h $PM_{2.5}$ forecasts from control and data assimilation experiments over the 10 EPA regions are discussed in the revised section.

(12) Lines 531-532 – Mention some statistics to demonstrate improvement.
Following another reviewer's recommendation, this section has been removed in the revised manuscript to improve clarity and conciseness. The discussion of data assimilation performance has instead been incorporated into an expanded Section 4 (formerly Section 4.1), which now provides a more comprehensive description of the results.

Minor comments:
(1) Line 22 – particles with aerodynamic diameters
Done.

(2) Line 40-41 – "PM2.5 is a major contributor to poor air quality in US" - Provide some references to this line.
Added references. Also replaced 'major' with 'key' in the revised manuscript.

(3) Line 44 – Kindly avoid the usage of term 'primary pollutants' in this context.
We replaced 'primary' with 'important' in the revised manuscript.

(4) Line 76 - replacing the previous offline-coupled the Global Forecast System
Done.

(5) Line 77-78 – "modeling system" repeated twice.

Fixed.

(6) Line 88 – Put 2.5 here (and in other places in manuscript) as subscript
Fixed.

(7) Line 170 – Check the sentence. Should it be: It is noted that a recipe that uses temperature…
The sentence is rephased to:
It is noted that a recipe to derive dry air density from temperature, surface pressure, and delta pressure has been added to VADER for cases where the dry air density is not provided.

(8) Line 203 - Mmountain Rregion?
Done.

(9) Line 213 – Add the last access date.
Done.

(10) Line 242 – and Texas.
Done.

(11) Lines 309-312 – Reduce the sentence length. Break into two sentences for ease of read.

We reduced the sentence length by breaking it into two sentences for easier reading. The sentence has been rephased as fellows:

 "Using the innovations and observation errors from subsection 2.2 and 3.2.3 as inputs to Equation 9, the background error variance of $PM_{2.5}$ was first estimated. This error, along with the background values, was then used in Equation 8 to estimate the scaling factor s. "

(12) Lines 345-346 – Is it "the ratio of the particle number concentration to total particulate volume"? Particle number concentration is already number per unit volume.

We agree that particle number concentration is already expressed per unit volume. However, we are not introducing a new variable; this simply states that the ratio of these two variables in the analysis is assumed to be the same as in the background. This allows the particle number concentration to be updated in the analysis according to the updated total particulate volume after data assimilation.

(13) Line 422 - Figure 6 presents shows time series of
Fixed.

(14) Line 580 – September 2025 or 2020?
Sorry for the typo. It is September 2020.

(15) Line 593 – There are two full stops before and after the reference.
Fixed.

---

## Author Comment (AC2)

General comments:

This study introduces assimilating $PM_{2.5}$ observations from AirNow and PurpleAir networks within JEDI for AQMv7, and analysized the assimilation benefits by conducting several experiments. The work is important and within the scope of GMD, and the manuscript is well-structured overall. However, there are still some concerns require to be addressed to further improve the quality of the manuscript before its publication.

We thank the reviewer for the thorough evaluation of our manuscript!

In response to your comments, we have revised the manuscript accordingly and addressed the issues raised to strengthen the clarity and robustness of the study. Below, we provide point-by-point responses to the reviewer's comments, with **our replies shown in blue font. The changes in the revised manuscript are also shown in the blue font.**

Major comments:

(1) Data assimilation of PM2.5 observations has been extensively studied. This study demonstrates the value of assimilating novel observations from PurpleAir for improving numerical air quality predictions. But the unique features or advantages of the PurpleAir network have not been fully demonstrated. It is suggested that the authors further strengthen the explanation of the novelty of this study.

Thank you for the constructive comments and suggestions.

A paragraph has been revised and expanded to summarize the novelty of this work. It is provided below (see also the second-to-last paragraph in the Introduction):

"This study aims to develop and evaluate an initial aerosol analysis capability for the AQMv7 system by assimilating $PM_{2.5}$ observations using the JEDI 3D-Var framework. Compared to previous $PM_{2.5}$ data assimilation studies, this research adopts the NOAA's regional operational AQMv7 system and incorporates a new $PM_{2.5}$ transform in JEDI for assimilating $PM_{2.5}$ observations. In addition to evaluating the impact of assimilating AirNow $PM_{2.5}$ measurements on air quality prediction, this study also examines the impact of assimilating low-cost PurpleAir observations. Although PurpleAir data are valuable for real-time air quality monitoring, their impact on numerical air quality prediction has not been thoroughly investigated. To the authors' best knowledge, this is the first study to demonstrate the value of PurpleAir observations for air quality prediction during a major fire event using the AQMv7 system."

Minor comments:

(1) The authors cited several webpages such as in Lines 58, 106, and 394 of the manuscript. It should be noted that web-based references have stability risks due to periodic or irregular maintenance of websites. If a cited page becomes inaccessible, for example the EPA webpage in L394, readers would be unable to verify the definition of "Regions 1–10", leading to confusion in subsequent region-based statistical analyses in the manuscript. Therefore, it is recommended that essential contextual information be incorporated in the manuscript to enhance completeness and readability. If the authors deem webpage citations necessary, the citation format should be revised to align with the requirement of the journal.

We agree with the reviewer's comments. We have minimized the use of webpage references where possible and incorporated essential contextual information directly into the manuscript to improve completeness and readability. Specifically, EPA regions are now defined in the main text where they are used, or the statements have been rephrased into more general statements when appropriate. For cases where webpage citations remain necessary, the citation format has been revised to comply with the journal's requirements.

(2) Figure8 (middle row, right panel) shows an increase in MAE over the eastern U.S. Does it indicate limitations of the PA data or its assimilation in these regions during the study period, and why.

Both initial condition errors and model errors can lead to the increased MAE in 24h forecast in the PurpleAir data assimilation experiment over the eastern U.S. However, the current results indicate that this increase may be related to lower PurpleAir data quality in this region. This is supported by the presence of elevated MAE at several stations at the 1-h forecast time, which is close to the analysis time. In contrast, the MAE increase at the 24-h forecast lead time is much less pronounced in the experiment in which only AirNow $PM_{2.5}$ observations were assimilated, further suggesting that the issue is associated with PurpleAir data rather than model forecast error growth. A brief discussion on this was added in the revised manuscript in Section 4.

(3) Figure8: The colorbar ranges in the top and bottom row of the left panel looks truncated. It is suggested to expand the colorbar range to fully capture the extent of MAE reduction. Meanwhile, it is recommended that each panel within multi-panel figures be labeled with letters (a, b, c, etc.) to facilitate clear referencing in the manuscript text.

The color bar is chosen to clearly show the impact of PurpleAir data at 1h forecast time.

As described in the reply to the previous comments, as part of this revision, the figures were removed, but a table of statistics over different areas were added to clearly show the data impact.

(4) In Lines 88, 129, 211, 217: The "2.5" in "PM2.5" should be formatted as subscript to follow the convention and keep consistent with the others in the manuscript.

Fixed.

---

## Author Comment (AC3)

The manuscript titled " PM2.5 Assimilation within JEDI for NOAA's Regional Air Quality Model (AQMv7): Application to the September 2020 Western U.S. Wildfires" by Wang et al. with reference egusphere-2025-4098 is a valuable scientific contribution in operational aerosol assimilation, particularly for wildfire smoke events. Overall, it is very well written and provides a detailed investigation of the PM2.5 assimilation of AirNow and PurpleAir surface observations in AQMv7 for the US in September 2020. The spatio-temporal investigation of the influence on different forest lead times (1-24hours) provide interesting insights.

Overall, I suggest accepting the manuscript with the following major corrections:

Thank you for the thorough evaluation of our manuscript!

In response to the reviewer's comments, we have revised the manuscript accordingly and addressed the issues raised to strengthen the clarity and robustness of the study. Below, we provide point-by-point responses to the reviewer's comments, with **our replies shown in blue font. The changes in the revised manuscript are also shown in the blue font.**

**General comments:**

A) My first general comment is on the specific purpose of this paper: For me, it is not clear whether the novelty of the contribution is about (1) the general implementation of an PM2.5 assimilation system for surface stations, or (2) just about the impact of additionally assimilating PurpleAir observations.

Thank you for the comment.

This research adds $PM_{2.5}$ data assimilation capability for the NOAA operational AQMv7 model within the JEDI framework and evaluates the impact of assimilating $PM_{2.5}$ observations, including PurpleAir data, on the prediction of the September 2020 western U.S. wildfire event. In the revised manuscript, we have revised and expanded a paragraph to more clearly summarize the novelty of this work (see the second-to-last paragraph of the Introduction). The revised text is provided below.

Revised manuscript text:

"This study aims to develop and evaluate an initial aerosol analysis capability for the AQMv7 system by assimilating $PM_{2.5}$ observations using the JEDI 3D-Var framework. Compared to previous $PM_{2.5}$ data assimilation studies, this research adopts the NOAA's regional operational AQMv7 system and incorporates a new $PM_{2.5}$ transform in JEDI for assimilating $PM_{2.5}$ observations. In addition to evaluating the impact of assimilating AirNow $PM_{2.5}$ measurements on air quality prediction, this study also examines the impact of assimilating low-cost PurpleAir observations. Although PurpleAir data are valuable for real-time air quality monitoring, their impact on numerical air quality prediction has not been thoroughly investigated. To the authors' best knowledge, this is the first study to demonstrate the value of PurpleAir observations for air quality prediction during a major fire event using the AQMv7 system."

While the text (title, abstract, introduction) suggest (1), some parts remain confusing in this regard:

1. In l. 81-84 you mention other literature assimilating AirNow PM2.5 in the US. Please specify the distinctive novelty of this contribution compared to previous work.

We have revised and extended a paragraph to summarize the novelty of this work. Please see the previous reply.

2. It is not clear in Sec.2 what the new developments are and what was already available before. Please state more clearly in the beginning of the section or for each subsection.
▪ incl l.129: is this part of the new developments?, otherwise please add reference

This section provides a general description of the 3D-Var method used in this research. The $PM_{2.5}$ assimilation capability builds upon the existing JEDI framework rather than constituting a new methodological development. Accordingly, we have added an appropriate reference to JEDI to clarify this point.

▪ incl l.159ff: (same)

Section 2.2.1 describes the new developments introduced for $PM_{2.5}$ data assimilation in this study. We first describe $PM_{2.5}$ is calculated within the AQMv7 model, then present the development of $PM_{2.5}$ transform within JEDI for data assimilation. To make this clearer, the revised manuscript explicitly states that "The $PM_{2.5}$ observation operator is constructed by combining the newly developed $PM_{2.5}$ transformation recipes in the JEDI Variable Derivation Repository (VADER) with an existing general spatial interpolation operator in the Unified Forward Operator (UFO)." Other development includes a new recipe to derive dry air density from temperature, surface pressure, and delta pressure.

3. Along these lines, it is also not clear in the summary in Sec.5 if l. 565-572 is about new developments. Please specifically summarize the purpose and novel developments of this study in Sec.5 (see also general comment D).

Alone with the new development above, other new applications include the specification of background error variances, and updates of aerosol particle number concentrations.

B) Sec. 4.1 contains plenty of plots which are only discussed shortly. This is fine, but I would suggest moving some less important plots in the Apx. (e.g. Fig.5, Fig.8) to increase readability. Or divide into subsections and consider expanding the description of results from each plot (eg temporal, spatial comparison, especially if Sec.4.2 is deleted/shortened:
see general comment C)

Thank you for the constructive comments. We have restructured Section 4 by removing section 4.2. The revised manuscript provides a clearer and more detailed interpretation of the results shown in the remaining figures and added new figures and discussion.

C) Sec. 4.2: Concerning the whole content of Sec.4.2: The only new result I can see in this section is l.538-541. All the rest seems to reproduce the results of Sec.4.1. I suggest removing this whole section, and maybe putting some of the plots in the Apx. (eg Fig.10 if the results in l.538-541 are considered important). Otherwise:
∘ Please explain more clearly why you are specifically looking at the forecast initialized at 12UTC. You mention the operational 72-hour forecast, but only show results for forecast

hour 1-24. What additional information does this investigation provide compared to the
overall results above?
◦ e.g. Given the fact that I don't know where Areas B and C are exactly (see smaller
comment 15): What is the additional result of l.531f compared to Fig.7? Also l.533-537,
l.542-544
◦ Please restrict this section to specifying and discussing the novel results that cannot be
obtained from Sec.4.1.

Thank you for your insightful comments! We agree that removing the section 4.2 does not affect the
overall integrity and main conclusions of the manuscript. In the revised manuscript, the original section
4.2 was removed. However, a new figure was added to demonstrate the data assimilation impact on EPA
regions.

D) Sec.5: I'm missing an actual summary and discussion of the results of this study. What conclusions do
you draw from the results above? You show plenty of plots with different aspects in Sec.4, I would
wonder if the only important aspects you want to summarize and discuss are in l.573-584.

In addition to addressing the comment A.3 that is to summarize the purpose and novel developments of
this study in Sec.5, we briefly summarize the specification of background error variances, updates of
aerosol particle number concentrations, and data assimilation impact on forecast.

1. e.g. Do you have any explanation or conclusion from the temporal and spatial differences between the
runs that you describe in Sec. 4?
Overall, assimilating AirNow data alone or together with PurpleAir data, improves forecast skill (as measured by
MAE and RMSE) for up to 24 hours. Reductions in MAE and RMSE are observed across all EPA regions, with the
largest improvements occurring in areas affected by fire events or influenced by transported smoke. These results
suggest that data assimilation is most valuable in regions with highly variable pollutant levels, and primarily
enhances short-term forecast accuracy.

2. The remaining parts of Sec.5 are about previous work (as far as I understand, l.565-572, compare
general comment A:3) and outlook (l.585-608).
You are correct. We had shortened the above contents in the revised manuscript.

**Smaller comments:**
1. Sec.3 includes lots of different subsections. I would suggest giving a short overview of the
content at the beginning of the Sec.3.
Thanks for your suggestion. A short overview was added at the beginning of Sec.3.
Revised manuscript text:
"In this section, the September 2020 western U.S. wildfire event and the model configuration are first
briefly introduced. This is followed by a description of the $PM_{2.5}$ observation processing, including
PurpleAir $PM_{2.5}$ quality control, bias correction, and observation error specification. Next, the background
error specification and the updates to total particle number and surface area concentrations are presented.
Finally, the design of the data assimilation experiments is described. "

2. Sec.3.1: Please provide some more general information on the region that you're discussing,

eg west/east? which state?... For a non-US reader it's not obvious where the Willamette Valley is. And please specify what you are referring to as "broader region"?

Thanks for pointing this out. We have added that the Willamette Valley is located in western Oregon. Additionally, we clarify that the "broader region" refers to the Pacific Northwest, including Oregon and Washington. These changes provide context for non-U.S. readers.

3. l. 234f: Which reference PM2.5 data did you use in your modified regression equation?
The reference data is AirNow PM data. The hourly regression equation is removed since it is not actually used.

Please specify.
4. l. 244-256: Please restructure this paragraph. If I understand correctly, you're switching between AirNow (l.245, l.247f, l.250ff) and PurpleAir (l.246, l.248f) multiple times. Please formulate more clearly.
∘ l. 247f: Does this refer to PurpleAir? Please specify.
∘ l. 248f: Redundant information with l.245.
∘ l.249 "5%": For comparison to your setup, please specify the type of observations they assigned with 5% errors, eg lowcost or "AirNow-like" stations?
The paragraph was restructured. It explicitly states "The AirNow PM$_{2.5}$ observation errors were set to 5% of the observed values. For PurpleAir PM$_{2.5}$ data, the observation errors were set to 10%".

∘ l. 253: Unit of "1.5" missing (should be the same as PM2.5 concentrations, no?)
Contents removed.

∘ l. 254ff: Unclear, please specify where these numbers coming from. Is this a result from the spatial averaging to a 0.1deg grid? As far I understand, this can only reduce the effect of large PM2.5 observations if there are always multiple observations within one gridcell. Is this always the case? Please explain.
Contents removed.

5. l. 259: Please specify the matched stations in Fig.1d. Does this refer to locations where both networks have closely located stations? If yes, which criteria did you use to define "closely located"?
∘ At the end of this subsection, it is also not clear why you are looking at the matched values in Fig.1d. What does it tell you? Is there any consequence you are taking from it?
Otherwise, remove Fig.1d.
Figure 1d was removed.

6. l. 289: Is the linear relationship the same for all 70 variables? Please describe.
Yes. This is explicitly stated in the manuscript.

7. l.308: inconsistent notation of innovation vector: here(l.308) bold d, Eq.2: non-bold d, Eq.9: non-bold d with subscript b and superscript o. Please unify or define differences.
The inconsistent notation was fixed.

8. l. 338: I would associate a "cutoff scale" this to be the distance at which the correlation function is "cut off", assuming zero correlation beyond that. But this is different from a correlation length that determines the shape of the correlation function itself. Please specify.

You are correct that a "cutoff scale" is the distance at which the correlation function is "cut off", assuming zero correlation beyond that. JEDI uses these cut-off scales in background error correlation parameters, which are different from GSI and WRFDA.

9. l. 374: Please specify what happens to the PM2.5 field at the initialization times. Do you initialize from the last analysis? Since you assimilation PM2.5 every 3 hours, what is the difference in PM2.5 between the assimilation times and the initialization times? Or does the initialization only apply to meteorological fields?

At each analysis time, all aerosol-related prognostic variables are updated through data assimilation. The updated aerosol fields, together with the meteorological initial conditions, are then used to initialize the subsequent forecasts. The experiments are fully cycling data assimilation and forecasting experiments; therefore, except for the very first forecast, all forecasts are initialized from the aerosol analysis of the previous cycle. Data assimilation is performed every 3 hours, and a 3-hour forecast is launched at each cycle. This 3-hour forecast serves as the background for the subsequent data assimilation cycle. In addition, forecasts initialized at 0000, 0600, 1200, and 1800 UTC are extended to 24 hours for evaluation purposes. The above description is incorporated in the revised manuscript.

Please see Section 3.1 and 3.5 for more information in the revised manuscript.

10. l. 383: Are these independent observations left out for the experiments that assimilate AirNow? And if not: How can the bias of DA_AN be worse than DA_ANPA (and DA_PA) during the first forecast hours, if compared agianst the same assimilated AirNow data (Fig.3)? Please discuss the implications of validating the different experiments with AirNow data concerning potentially temporally-correlated errors.

No independent AirNow observations were withheld for evaluation.

In the plot, bias is shown as a function of forecast lead time for all forecasts initialized four times daily. When examining the 1-hour forecast bias for each individual cycle (00Z, 06Z, 12Z, and 18Z, Table. s1), the DA_AN experiment exhibits the best performance in terms of bias except for the cycle at 06Z. When all forecast samples are combined, the positive and negative biases in the DA_PA and DA_ANPA experiments largely cancel out, resulting in a near-zero overall bias. That is why we also need to look at MAE and RMSE. It is seen in the below Table. s1, the DA_AN experiment consistently exhibits the best performance in terms of MAE and RMSE.

**Table s1.** Bias, MAE, and RMSE of forecasts initialized at 00Z, 06Z, 12Z, and 18Z, evaluated at forecast hour 1.

| INIT Time | EXP | BIAS | MAE | RMSE |
|-----------|-----|------|-----|------|
|           | CTR | -4.6 | 14.9 | 52.7 |

| | | | | |
|---|---|---|---|---|
| 00Z | DA_AN | 0.1 | 4.1 | 15.3 |
| | DA_PA | -1.9 | 10.0 | 33.0 |
| | DA_ANPA | -0.9 | 5.0 | 19.7 |
| 06Z | CTR | -1.1 | 16.3 | 66.8 |
| | DA_AN | 0.7 | 4.7 | 23.1 |
| | DA_PA | -0.2 | 10.6 | 41.8 |
| | DA_ANPA | 0.2 | 5.7 | 27.4 |
| 12Z | CTR | 3.3 | 18.5 | 112.3 |
| | DA_AN | 1.7 | 5.5 | 31.3 |
| | DA_PA | 2.5 | 12.8 | 89.3 |
| | DA_ANPA | 1.6 | 6.7 | 38.7 |
| 18Z | CTR | -1.6 | 15.9 | 65.8 |
| | DA_AN | 0.3 | 4.5 | 16.7 |
| | DA_PA | -0.7 | 10.5 | 42.3 |
| | DA_ANPA | -0.6 | 5.3 | 19.2 |

Moverover, the control run appears to overshoot the diurnal cycle, with a negative bias of −4.6 at 00Z and a positive bias of 3.3 at 12Z (Table. s1). These relatively strong systematic biases may limit the effectiveness of data assimilation compared to cases with weaker bias since data assimilation theory usually assumes an unbiased prior (background) state.

It is noted that in theory, data assimilation minimizes the weighted squared difference between the model state and observations, but it does not explicitly guarantee a reduction in bias. While bias is often reduced in practice, this is not always the case. Moreover, the calculated bias does not account for the relative weights assigned to individual observations in the assimilation process in this study.

We acknowledge that validating experiments against assimilated AirNow observations may introduce temporally correlated errors, particularly at short lead times.

11. Fig.3: Please discuss why the bias of the joint assimilation DA_ANPA is closer to CTR than each of the single-obs. assimilations DA_AN and DA_PA. Please also discuss the change in biases over time, i.e. increasing underestimation with forecast time which makes the single-obs. assimilations DA_AN,DA_PA being least biased for longer forecast hours.

In the revised manuscript, we discussed the bias at the forecast hour 1 and the trend of bias.

As shown in the reply to the previous comment, *bias* is limited because negative and positive biases cancel each other out. That is why we need to look at skills in terms of MAE and RMSE.
The bias in the control run follows an upward trend initially, then reverses into a downward trend. The data assimilation runs exhibit a similar trend, since data assimilation generally adjusts the initial state variables rather than eliminating systematic model bias.

This paper primarily focuses on the development of the data assimilation scheme and the evaluation of data impacts. We think that the observed bias evolution is mainly controlled by diurnal cycles/variations but dominated by certain forecast cycles (for example, 00Z) and their interactions between fire emissions and meteorological processes, particularly boundary layer physics (e.g., the diurnal evolution of PBL height).

12. l. 458f: This can hardly be quantified from Fig7. For the assimilation runs, it would be helpful to show their differences to AirNow validation observations. I would suggest e.g. removing their absolute PM2.5 fields (Fig.7c,e,g) and showing their differences to AirNow validation along with their differences to the control (keeping Fig.7,d,f,h).

Thank you for the comment.
One unique feature of this fire event is the extremely high PM2.5 values, ranging from 0 to 800 µg m$^{-3}$, which makes it difficult to visually differentiate differences between experiments over the large CONUS domain. We explored various approaches, including the one you suggested, but ultimately chose the current figures because they best serve the goal of our presentation.

The purpose of these panels is to illustrate how the 1-h forecasts from the data assimilation experiments are improved relative to the control (CTR) run. For this reason, the CTR experiment is used as the reference in Figures 7d, 7f, and 7h, highlighting the relative impact of data assimilation. By comparing these panels with Figure 7b, the improvements in the 1-h forecasts from the assimilation experiments can be more clearly identified.

13. l. 483ff: I assume you refer to mean reductions (domain-averaged)? Local reductions seem to be much larger and varying. Moreover, when you only discuss mean reductions, this does not fit to in the discussion on spatial distributions here (introduced in l.442f, summarized in l.487ff). Please either discuss spatial distributions OR
◦ show only mean reductions in the figure (replacing Fig.8)
◦ and move this paragraph where it fits
You are correct that the mean reductions shown in Fig. 8 represent domain-averaged values over all stations, while local reductions can be larger and more variable.
To address your comment, we have restructured this section to focus on the main results. Figures were updated to show only the 24-h MAE change for clarity. To also capture spatial variations, a new figure presenting statistics across EPA regions 1–10 has been added, clearly illustrating the impact of data assimilation regionally.

14. l. 491f: This was not mentioned above. From which plot do you see this? Please specify.
Figure. 7.

15. l. 517-522: Please write a bit more where these Areas A,B,C are located (eg states). For external readers, it's hard to guess just from the given information.
Related contents are removed in the revised manuscript.

16. l. 525f: Is this a guess? Or how do you conclude this?
Related contents are removed in the revised manuscript.

17. l. 579-582: The sentence suggests that these results from the given reference. But the reference is from 2023 while the case study is from 2025, that's not possible. Please clarify.
Sorry for the typo about the year for the September fire event, which took place in 2020.

18. l. 582f: Now it's November 2025, I assume the data coverage did not change significantly within 2 months. Furthermore, you only use data from 2020 in this study. Is there maybe a mistake in the year (see also comment above)? Or I completely misunderstand this paragraph.
Thank you for pointing this out. The year has been corrected in the revised manuscript.

19. l. 585-595: Confusing. You talked about the results before (l. 573-584), and now afterwards about the implementation (l.585-595). I suggest restructuring.The year has been corrected in the revised manuscript.
The paragraph was rephased to reduce confusion. The year had been correct as well.

20. l. 596-608: I'm not sure if you need to discuss these challenges in that much detail here. This work is about assimilation of surface observations, and in my point of view it's enough here to say that you are planning to include improved AOD assimilation into the system.
The details on challenging in assimilating AOD are removed in revised manuscript.

Technical and formulation-related comments:
• l. 27-29: That's a very technical sentence for the abstract. At this point, it is not clear what control variables are in this context. And how background error standard deviations scale to background state values. It looks like quite a lot of information was squeezed into one sentence. Is it necessary to be included in the abstract? If yes, please expand, otherwise I suggest removing here.
Thanks for your suggestion. A key component in a data assimilation scheme is the control variable and their background error statistics. Limited to the length of the abstract, we don't explicitly list the names of all the 70 control variables, which are prognostic variables in the AQMv7.0 model. However, it is clearly mentioned that the control variables are individual aerosol species, how their errors are generally specified. Details can be found in section 3.

• l. 34: Please explain the abbreviation "CONUS" once.

Updated to Continental United States.

• l. 45f: Do you refer to the specific AQI of EPA, or in general to any AQI? Please specify.
It is the specific AQI of the U.S. Environmental Protection Agency. The manuscript is updated to reflect this.

• l. 179-183: Is there a square missing somewhere? The diagonal of a covariance is the variance, squared standard deviation. So Sigma should be the variance matrix, or it has to be Sigma^2 in Eq.(5). (also in l.184ff).
The error covariance B is decomposed into a standard deviation matrix ($\Sigma$) and a correlation matrix (C), $B=\Sigma C\Sigma$
$\Sigma$ is a diagonal matrix, with the standard deviation. If the correlation is ignored, $B=\Sigma\Sigma$, which is the multiplication of two matrices. The results will be the square of standard deviation as you mentioned.

• l. 198f: please specify. In that region (which region, see smaller comment 2)? Or in whole US?
The region is the Pacific Northwest (primarily eastern Washington and western Oregon) during the September 2020 wildfires. The manuscript is updated accordingly.

• l. 240ff: Can this be seen from Fig.1b? Please add reference to plot.
The PurpleAir coverage (Fig. 1b) was well described in Section 3.2.3. So that the related description was removed in Section 3.2.2.

• l. 257f: The two sentences introducing different plots in Fig.1a-b and Fig1.c-d are confusing, because they are disconnected from their descriptions/interpretations below. I suggest moving the sentences referring to Fig.1a-b (l.259-263) directly after mentioning Fig.1a-b here. And moving the sentence introducing Fig.1c-d down right before its description in L.263ff.
The paragraph was rephrased according to your comments.

• l. 297-301: You are mentioning multiple times that background PM2.5 error variance is denoted as Sigma^2. That might be confusing. I'd suggest defining once, and using either words or symbol afterwards.
We reduce use of the symbol and keep it only when describing the Eq. 9.

• l. 327: Technically, Fig.2 is not not PM2.5 space. This would be eg showing the background error standard deviation as function of PM2.5. Do you mean the PM2.5 background error standard deviation as (weighted) sum over all 70 variables?

The figure is intended to illustrate the key difference between the proposed dynamically location- and time-dependent varying background error formulation and the traditional static, constant background errors. Rather than displaying background error standard deviations for individual aerosol species, Figure 2 shows the effective $PM_{2.5}$ background error standard deviation after applying the scaling factor. This $PM_{2.5}$ error standard deviation reflects the combined contribution of aerosol species to $PM_{2.5}$, rather than a simple weighted sum of all individual prognostic variables.

• l. 389f: This sentence is doubled with l.383.
Fixed.

• Fig.4: Description of mean missing in figure caption.
Fixed.

• l. 422: two verbs ("presents shows")
Fixed.

• Fig.5, 7, 8, ...: Icons and labels are very small and hard to see/read.
Fig. 5 has been removed in the revised manuscript. Figures 7 and 8 have been slightly adjusted. We can further enhance them if necessary.

• l. 523: capital "F" in "figure"
Fixed.

• l. 593: remove "." before reference
Fixed.

• l. 593ff: is an ensemble prediction system planned for AQM? Please explain or remove
Sentence.
Removed.